# CURSE OF INSTRUCTIONS: LARGE LANGUAGE MODELS CANNOT FOLLOW MULTIPLE INSTRUCTIONS AT ONCE

## ABSTRACT

Large language models (LLMs) have demonstrated impressive performance across various natural language processing (NLP) tasks owing to the strong capability of following instructions. To further accelerate the integration of LLMs into our society, it is essential to have LLMs follow many instructions as accurately as humans do. This study reveals that LLMs unexpectedly struggle to follow all instructions simultaneously as the number of instructions increases. First, to validate our claim, we introduce ManyIFEval, a large-scale benchmark dataset comprising task prompts with up to ten objectively verifiable instructions. Second, we conduct experiments based on ManyIFEval with GPT-4o, Claude-3.5, Gemini-1.5, Gemma2, and Llama3.1, demonstrating that as the instruction count rises, the models' ability to follow individual instruction deteriorates gradually but constantly. As a result, the models' ability to follow all the instructions significantly drops: the success rate of all the instructions is precisely explained by the success rate of individual instructions to the power of total number of instructions. We refer to it as the "curse of instructions". Third, to remove the curse without retraining models, we propose an inference-time strategy that enhances performance through iterative self-refinement. We demonstrate that instruction-level chain-of-thought reasoning significantly improves their capability to detect and correct instruction-following errors. Notably, our method has improved the success rate of following ten instructions by GPT-4o from 15% to 31% and Claude 3.5 Sonnet from 44% to 58%. We also show that precision is more important than recall in feedback: just telling LLMs that they are not following all the instructions also improves self-refinement success. Our findings highlight a fundamental limitation of instruction-following ability and suggest a future direction for building trustworthy LLMs that can coexist with human society. [1]

## 1 INTRODUCTION

Large language models (LLMs) have demonstrated impressive performance across various natural language processing tasks (OpenAI, 2024b; Google, 2023). When utilizing LLMs for solving diverse tasks, prompting (Brown et al., 2020; Vatsal & Dubey, 2024; Sahoo et al., 2024) has become a prevalent approach. Prompting involves providing textual instructions to guide the model's behavior towards desired outcomes. This instruction-following ability mainly comes from instruction tuning process (Wei et al., 2022a; Zhang et al., 2023; Chung et al., 2024).

As LLMs continue to evolve, their ability to follow multiple instructions simultaneously becomes crucial for broader applications. In human society, individuals perform activity under numerous laws and organizational norms. For LLMs to be more reliable and widely applicable in human contexts, they must adhere to the various rules. Moreover, in domain-specific applications, LLMs need to comply with established guidelines and regulations. For instance, legal applications require adherence to thousands of statutes (Guha et al., 2023), while medical applications must follow numerous clinical guidelines (Singhal et al., 2023; Kasai et al., 2023; Hager et al., 2024).

---

[1]ManyIFEval dataset and inference code are available at `https://anonymous.4open.science/r/ManyIFEval`

Figure 1: Problem of curse of instructions.

However, the extent to which models can adhere to multiple instructions simultaneously remains unclear. To address this question, we construct ManyIFEval, a large-scale benchmark dataset comprising task prompts with up to ten objectively verifiable sets of instructions. These instructions are designed with a difficulty level such that major models can successfully follow them in most cases. Unlike existing benchmarks that typically provide only a few instructions at a time (Wang et al., 2022b; Wei et al., 2022a; Conover et al., 2023), our benchmark scales up to ten concurrent instructions by extending IFEval (Zhou et al., 2023) .

Our experiments reveal that as the number of instructions increases, models struggle to adhere to all of them by following a simple rule of instructions. Specifically, we conduct experiments based on ManyIFEval with various type of models, including closed models (GPT-4o (OpenAI, 2024a), Claude 3.5 Sonnet(Anthropic, 2024), and Gemini 1.5 Pro (Gemini Team, 2024)) and open models (Gemma2 (Gemma Team, 2024) and Llama3.1 (Llama Team, 2024)). The experiments demonstrate that as the instruction count rises, the models' ability to follow individual instruction deteriorates gradually but constantly. Consequently, the models' ability to follow all the instructions significantly drops by following a simple rule: the success rate of all the instructions is precisely explained by the success rate of individual instructions to the power of total number of instructions. We refer to it as the "curse of instructions".

To alleviate the curse of significant performance drop without additional training, we explore inference-time strategies (Brown et al., 2024; Welleck et al., 2024) that enhance performance through iterative self-refinement (Madaan et al., 2023). We find that having the model refine its response by feedback from itself leads to improved performance. When creating this feedback, it is beneficial to use Chain-of-Thought reasoning (Wei et al., 2022b; Kojima et al., 2022) for each instruction. This approach enhances performance by increasing inference time without the need for model retraining, making it applicable to models available only through APIs and avoiding the substantial costs associated with retraining model weights. Notably, our method has improved the success rate of the following ten instructions by GPT-4o from 15% to 31% and Claude 3.5 Sonnet from 44% to 58%. We also show that recall is more important than precision in feedback: just telling LLMs that they are not following all the instructions also improves self-refinement success.

In summary, our contributions are as follows.

- We construct and release a new benchmark called ManyIFEval for measuring the performance of multiple instruction adherence by scaling the axis of instruction count.

- Comprehensive analysis based on ManyIFEval with various types of closed models (GPT-4o, Claude-3.5, and Gemini-1.5) and open models (Gemma2 and Llama3.1) demonstrate that even the state-of-the-art models struggle to follow multiple instructions at once when scaling the number of instructions.

- We reveal a secret principle behind the significant drop in performance: as instruction count rises, the models' ability to follow individual instruction deteriorates gradually but constantly. Consequently, the success rate of following all the instructions significantly drops

by following the success rate of individual instructions to the power of total number of instructions.

- We propose a method to mitigate the performance degradation by iterative self-refinement through self-feedback loops in combination with chain-of-thought reasoning for each instruction. We also show that recall is more important than precision for the feedback.

- Our findings highlight a fundamental limitation of instruction-following ability and suggest a future direction for building trustworthy LLMs that can coexist with human society.

## 2 RELATED WORKS

### 2.1 BENCHMARKS FOR EVALUATING MULTIPLE INSTRUCTION-FOLLOWING PERFORMANCE

Instruction-following is a critical capability of large language models (LLMs), and various benchmark datasets have been proposed to evaluate this aspect (Wang et al., 2022b; Li et al., 2023; Zheng et al., 2023; Hayati et al., 2024; He et al., 2024b; Jiang et al., 2024; He et al., 2024a). Our study focuses on the relationship between instruction-following performance and the complexity of instructions, specifically measured by the number of instructions within a prompt. While several benchmarks investigate this relationship from different angles, our work uniquely emphasizes the impact of instruction count on model performance.

For instance, FollowBench (Jiang et al., 2024) incrementally adds up to five constraints (e.g. Situation, Format, Style) to a single prompt, organizing data according to the number of added constraints. FollowBench also includes a subset called "Mixed Constraints," which, like our work, adds constraints of different types to simulate real-world use cases. However, their Mixed Constraints primarily involve only tasks that need to edit or transform input sentences. Their findings indicate that instruction-following performance deteriorates as the number of constraints increases. However, the experiment was conducted with a limited number of instruction count and sample size.

ComplexBench (He et al., 2024a) includes not only "And"-type instructions—independent instructions that the model must satisfy simultaneously, similar to our benchmark—but also "Chain" instructions that require sequential execution and "Selection" instructions that involve conditional branching. ComplexBench demonstrates that complex instructions, such as those in the "Selection" and "Chain" categories, pose significant challenges for LLMs in terms of compliance.

IFEval (Zhou et al., 2023), the basis of our ManyIFEval, consists of a set of 25 categories of verifiable instructions and 541 prompts to which 1 to 3 instructions are assigned. These prompts are first rephrased with few-shot prompting to increase the diversity of wording, and then manually checked to ensure that there are no conflicts between the instructions. Furthermore, the method of judging compliance is programmatic and deterministic, and analysis of the evaluation results leads to insights on instructions that are difficult to comply with and to comparisons among various LLMs.

Regarding the quality of the datasets, it is crutial to have a wide range of complexity and abundant samples for each number of instructions to verify the relationship between the increase in the number of instructions and LLM's ability to follow plural instructions. As shown in Table 1 and Figure 2, because FollowBench and IFEval have the small variation in the number of instructions, it can be difficult to adequately verify the capability of following multiple instructions of LLMs. Furthermore, although samples with 14 instructions are included in ComplexBench, such samples exceedingly rare and the majority of samples are concentrated within the range of 3 to 6, which means that the number of instructions is not balanced across samples. Contrary to them, ManyIFEval includes samples with up to 10 instructions and 100 samples are contained for each instruction count. Therefore, our benchmark is preferable because of the sufficient variation in the number of instructions and the balanced samples.

Regarding evaluation methods, approaches to assess instruction-following can be broadly categorized into human evaluation, model-based automatic evaluation, and program-based automatic evaluation. Human judgment is costly and time-consuming. While model-based evaluations (Zhang et al., 2020; Liu et al., 2023), which is partially adopted by FollowBench and ComplexBench, can be influenced by the evaluating model's biases and limitations (Wang et al., 2024; Shen et al., 2023; Zheng et al., 2023). Program-based methods, which evaluates instruction-following performance by programmable rules, offer a cost-effective and objective alternative, allowing for accurate assess-

Table 1: Feature comparison among multiple instruction benchmarks.

| Benchmark | Language | #Samples | #Prompts | #Instructions per sample (max) | Is the number of instructions per sample balanced? | Evaluation Method |
|---|---|---|---|---|---|---|
| FollowBench | English | 85 | 17 | ✗ 5 | ✓ Yes | ✗ Model & Program |
| IFEval | English | 541 | 541 | ✗ 3 | ✗ No | ✓ Program |
| ComplexBench | Chinese | 1150 | 1150 | ✓ 14 | ✗ No | ✗ Model & Program |
| **ManyIFEval** | English | 1000 | 100 | ✓ 10 | ✓ Yes | ✓ Program |

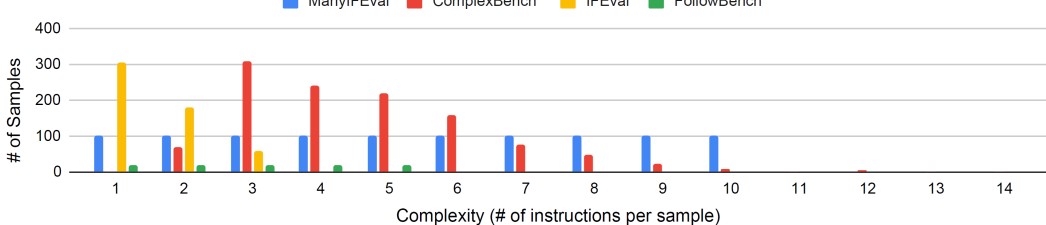

Figure 2: Histogram on the number of instructions per sample for each benchmark.

ments without the constraints of the other approaches. To facilitate reliable and efficient evaluation, we extend an existing dataset called IFEval, which allows for program-based automatic evaluation of instruction-following. By leveraging program-based evaluation, we create a benchmark that not only objectively measures LLMs' ability to follow complex instructions but also provides insights into how instruction count affects performance.

## 2.2 METHODS TO IMPROVE INSTRUCTION-FOLLOWING ABILITY

LLMs are primarily pre-trained on large corpora of web documents to predict subsequent words, without specific training to respond to human textual instructions. To enhance their ability to interpret and follow human instructions, instruction tuning (Wei et al., 2022a; Zhang et al., 2023) is performed as the most representative method for post pre-training. These methods train models on datasets consisting of instructions paired with appropriate responses, such as FLAN (Wei et al., 2022a), Alpaca (Taori et al., 2023), and Dolly (Conover et al., 2023).

More recent studies (Xu et al., 2023; He et al., 2024a) have highlighted that existing datasets often feature only few instructions for each task. The studies have synthesized more complex instruction datasets using LLMs themselves, reporting improvements in instruction-following performance through fine-tuning with these intricate data. While the target instruction count in the studies is limited, this perspective is crucial for enabling models to handle multiple instructions.

However, retraining LLMs is often impractical due to the substantial computational resources required and the risk of catastrophic forgetting, where retraining may degrade the model's performance on previously learned tasks. Additionally, for LLMs accessed via commercial APIs, users are often restricted to access to model parameters and to finetune LLMs.

Therefore, we explore performance enhancement methods that do not require retraining and are applicable to API-accessible models by focusing on inference strategies. Techniques such as generating multiple outputs and selecting the most suitable one through majority voting (Wang et al., 2022a), or self-refinement (Madaan et al., 2023), have demonstrated performance improvements across various tasks. This study focuses on self-refinement for improving the instruction following ability of multiple instructions, where LLMs first produce an initial response and execute a loop of error detection and refinement by themselves.

## 3 MANYIFEVAL

To investigate the relationship between the number of instructions and the instruction-following performance of language models, we constructed a dataset called ManyIFEval. This benchmark enables

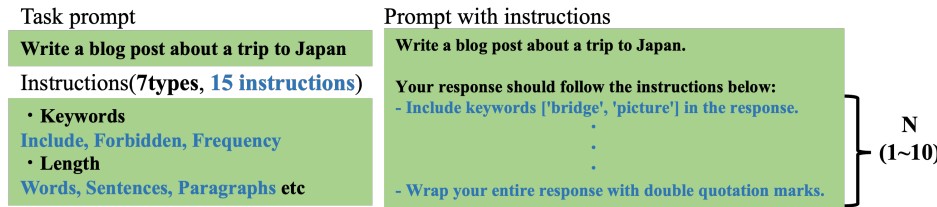

Figure 3: An example from the ManyIFEval dataset, a benchmark designed to evaluate how well models can follow up to 10 instructions simultaneously. Each example consists of a free-form task prompt, such as *"Write a blog post about a trip to Japan."*, accompanied by a set of objectively verifiable instructions that must be satisfied when performing the task.

an objective evaluation of how the increase in similar-difficulty instructions—up to a maximum of ten—affects a model's ability to follow instructions.

## 3.1 DATASET

As shown in figure 3, each instance in ManyIFEval consists of a free-form task prompt, such as *"Write a blog post about a trip to Japan."*, accompanied by a set of instructions that must be satisfied when performing the task. To systematically examine the impact of the number of instructions on performance, we created ten instances per task prompt, varying the number of instructions from one to ten.

Considering all possible combinations of which instructions to include and the order in which they appear would result in an enormous number of instances, leading to prohibitive inference costs during evaluation. Therefore, to maintain computational feasibility while ensuring a comprehensive assessment across various models, we limited the dataset to ten instances per task prompt, each with a different number of instructions ranging from one to ten.

The task prompts and instructions were extracted from the existing IFEval benchmark. In the extraction process, we selected task prompts that remain feasible even when additional instructions are provided, and instruction groups that are non-conflicting and can be simultaneously satisfied. Through manual verification, we curated a total of 216 task prompts from an initial set of 541. These were divided into 110 for training, 100 for testing, and 6 for few-shot prompting.

There are a total of 15 instructions categorized into the following six types:

- **Keywords**: Specifying keywords to include or exclude.
- **Length Constraints**: Specifying the length of the answer.
- **Detectable Format**: Requiring bullet point formatting.
- **Change Cases**: Specifying the use of uppercase or lowercase letters.
- **Start with / End with**: Instructions to begin or conclude with quotation mark.
- **Punctuation**: Prohibiting the use of commas.

Details of these instruction types are provided in Appendix B. For instruction types like *Change Cases* and *Length Constraints*, we select one variant each, ensuring that up to ten non-conflicting instructions are chosen from the total of 15.

The compliance with each instruction can be objectively verified programmatically, allowing us to determine success or failure in instruction following. Importantly, the selected 15 instructions are designed such that, when only one instruction is given with the task prompt, GPT-4o(gpt-4o-2024-05-13) achieves high success rate. GPT-4o achieves a success rate of over 90% for nine instructions, over 80% for four instructions, over 60% for two instructions. In other words, these instructions are designed with a difficulty level such that major models can successfully follow them in most cases: we excluded individually challenging instructions from our benchmark to focus on observing changes in performance as the number of instructions increases. By constructing the dataset with in-

structions of similar difficulty, ManyIFEval is well-suited for examining how increasing the number of instructions impacts a model's instruction-following performance.

## 3.2 EVALUATION METRICS

To evaluate instruction-following performance, we adopted metrics based on those used in IFEval and FollowBench. We define:

- **Prompt-level Accuracy**: The success rate of following all given instructions simultaneously for a particular prompt (equation 1). This assesses the model's capability to handle multiple instructions at once. This is Hard Satisfaction Rate (HSR) in FollowBench.

- **Instruction-level Accuracy (Inst-level accuracy)**: The success rate of following individual instructions in its response (equation 2). This metric assesses the model's ability to adhere to each instruction separately. This is Soft Satisfaction Rate (SSR) in FollowBench.

By applying these metrics, we can quantitatively assess how well language models perform as the number of instructions increases, providing valuable insights into their limitations and guiding future improvements in instruction-following capabilities. Concretely, Prompt-level accracy is defined as

$$\text{Prompt-level Accuracy (n)} = \frac{1}{m} \sum_{i=1}^{m} \prod_{j=1}^{n} s_i^j, \tag{1}$$

Where $m$ represents the number of prompts and $n$ represents the number of instructions per prompt, $s_i^j$ represents an binary metric if the target instruction $j \in n$ in a task $i \in m$ is sucessfully followed. $\prod_{j=1}^{n} s_i^j = 1$ if all instructions for prompt $i$ are satisfied, and 0 otherwise. Thus, this metric computes whether every instruction in a prompt is followed or not. Inst-level accracy is defined as

$$\text{Inst-level Accuracy (n)} = \frac{1}{mn} \sum_{i=1}^{m} \sum_{j=1}^{n} s_i^j, \tag{2}$$

where $s_i^j = 1$ if the $j$-th instruction of the $i$-th task is satisfied, and $s_i^j = 0$ otherwise.

## 4 EVALUATION

### 4.1 MODELS AND SETTINGS

To evaluate the impact of scaling the number of instructions on LLMs' ability to follow them, we conducted experiments using both closed-source and open-source models. For closed models which are accessible exclusively via APIs, we evaluated GPT-4o (gpt-4o-2024-05-13), Claude 3.5 Sonnet (claude-3-5-sonnet-20240620), and Gemini 1.5 Pro (gemini-1.5-pro-002). For open-source models, we assessed Gemma2 (gemma-2-9b-it) and Llama 3.1 (Meta-Llama-3.1-8B-Instruct). Inference for the open models was performed using the transformers (Wolf et al., 2020) and vLLM (Kwon et al., 2023) libraries. All models were evaluated using zero-shot prompting.

During decoding, we employed greedy decoding (top-k=1) to ensure deterministic outputs across models. However, for GPT-4o, we were unable to specify greedy decoding directly due to API limitations. As an alternative, we approximated greedy decoding by setting the nucleus sampling parameter to top-p=1e-10, effectively constraining the model to select the highest-probability tokens.

### 4.2 EFFECT OF INCREASING THE NUMBER OF INSTRUCTIONS

We investigated the effect of increasing the number of instructions on the models' instruction-following performance. In this experiment, we use a standard prompting method, zero-shot prompting: we set a task description and multiple instructions in a prompt and let the models generate an answer. Figure 1 (left) presents the inst-level accuracy and Figure 1 (right) presents the prompt-level accuracy and on the ManyIFEval benchmark. Our observations indicate a consistent trend across all models: even when individual instructions were based on simple rules, the models struggled to follow all instructions as the total number increased. And also, the results revealed that while the

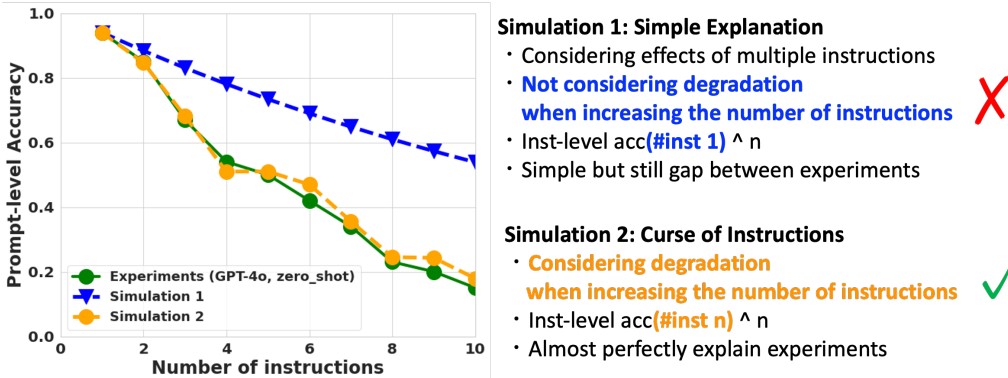

Figure 4: We empirically reveal a secret rule of LLMs when they follow multiple instructions: we refer to it as "curse of instructions". When given a total number of instructions to follow as $n$, LLMs obey a simple rule: Prompt-level accuracy (n) = (Instruction-level accuracy (n))$^n$. See Section 4.3 for the detailed explanation. See Appendix F for more results on various models.

models can follow instructions with high accuracy when presented individually, they experienced significant performance degradation of the same instructions as the number of accompanying instructions increased. This indicates that the presence of additional instructions adversely affects the models' ability to attend to and execute each instruction correctly.

### 4.3 CURSE OF INSTRUCTIONS

We empirically reveal a secret rule of LLMs when they follow multiple instructions: "curse of instructions". When given a total number of instructions to follow as $n$, LLMs obey a simple rule.

$$\text{Prompt-level accuracy (n)} = (\text{Instruction-level accuracy (n)})^n. \tag{3}$$

Recall that prompt-level accuracy and instruction-level accuracy are defined in Equation (1) and Equation (2) respectively. **This equation indicates that as the number of instructions increases, the overall instruction-following ability decreases endlessly, unless each instruction-following success rate is 100%.** However, unfortunately, we have confirmed that as the number of instructions increases, the models' ability to follow individual instruction deteriorates gradually but constantly (See Figure 1 (left)). This unacceptable fact further accelerates deterioration of the performance of prompt-level accuracy due to the accumulation of individual errors. As a result, the models' ability to follow all the instructions significantly drops. Figure 4 describes the simulation results of the rule. The result demonstrates that prompt-level accuracy is quite accurately explained by the above equation across models. See Appendix F for more results. In summary, the current LLMs inevitably cannot follow multiple instructions at once. This is an extremely inconvenient fact when we are implementing LLMs in human society, which always have many rules.

## 5 INFERENCE TIME ALGORITHMS TO MITIGATE THE PROBLEM

### 5.1 OUR METHOD

To alleviate the "curse of instructions", we explore a method to enhance instruction-following performance of LLMs without modifying their parameters. Given the enormous number of parameters in LLMs, fine-tuning or retraining is computationally expensive, and some models are accessible only via APIs, limiting the ability to adjust their internal weights. Therefore, we investigate inference-time strategies that can improve performance without altering the underlying model.

We propose a method of iterative self-refinement in which the model itself evaluates whether the model successfully follow each instruction and then corrects its answers based on this evaluation. An overview of the proposed method is illustrated in Figure 5. Specifically, given a task description and instructions to follow within a prompt, our method lets the model first generate an initial answer

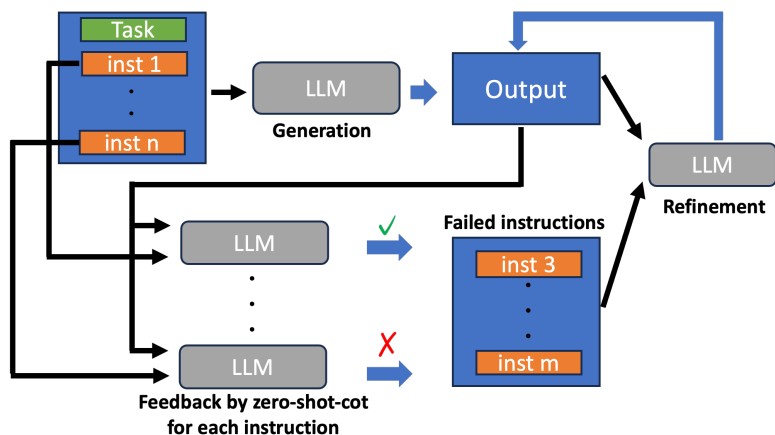

Figure 5: Overview of our method.

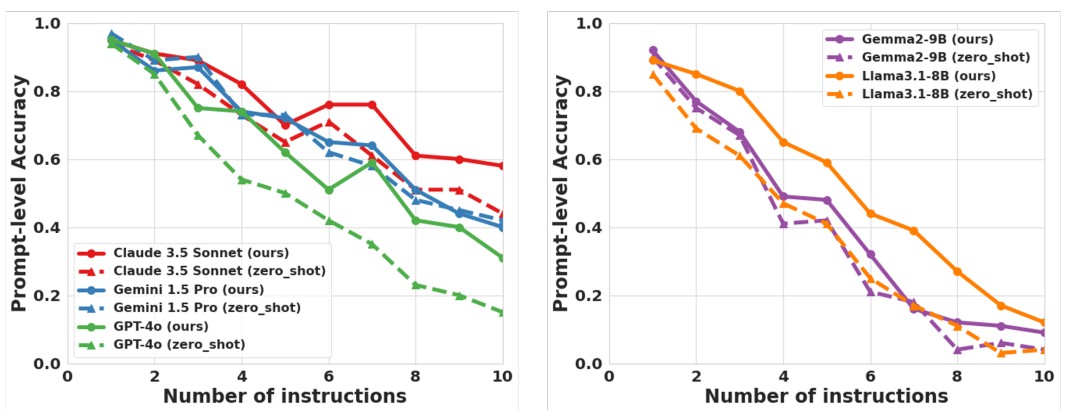

Figure 6: Performance improvement by our method.

(output). As a subsequent process, the model by itself judges the success or failure in following each instruction for the output text. Our method utilizes Zero-shot Chain-of-Thought (Zero-shot-CoT) (Kojima et al., 2022) to improve the judgement ability by enhancing the reasoning. Based on the hypothesis that attentions on multiple instructions may degrade feedback performance of LLMs, our method lets LLMs judges each instruction one by one rather than assessing all instructions at once. After the feedback, we set a subset of instructions that are judged as failure as well as the previous output into the prompt to let the model refine the output texts. Our method iteratively repeat the self-feedback and self-refinement cycle for a fixed number of times $T$. We set $T = 5$ across all the experiments.

The Experiment results of our method is presented in Figure 6. We observed that the performance of our method drastically improved compared with zero-shot, which is a standard way of just outputting an answer based on the instructions in a prompt (See Section 4.2). Notably, our method has improved the success rate of the following ten instructions by GPT-4o from 15% to 31%, Claude 3.5 Sonnet from 44% to 58%, and Llama3.1-8B from 4% to 12%.

## 5.2 COMPARISON WITH BASELINE METHODS

To show that our method is appropriately designed so that it achieves the better performance than naive implementation, we conduct a comparison experiment with several baselines which are created by decomposing technical elements in our method. The following is the list of comparison method. Figure 7 also describes the overall procedure of each method.

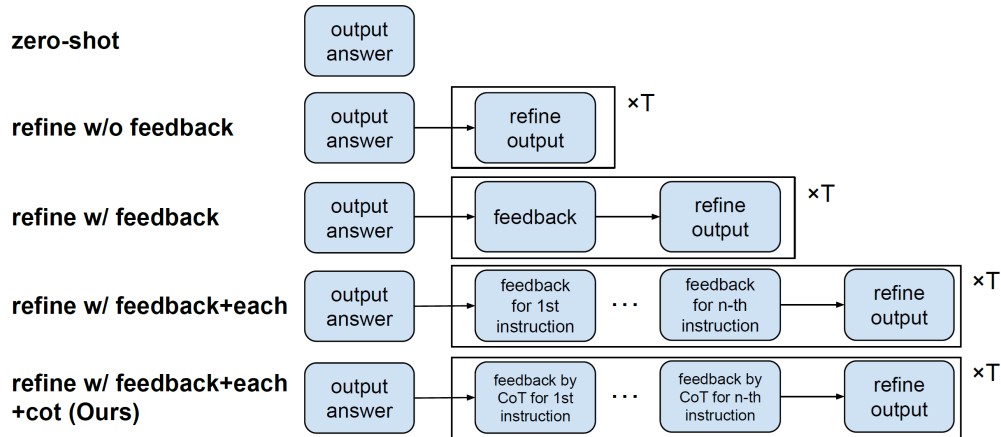

Figure 7: Method Comparison

- **zero-shot**: sets a task description and multiple instructions in a prompt and let the models generate an answer. See Section 4.2.

- **refinement w/o feedback**: self-refines the initial output $T$ times (a predetermined number of iterations) after the initial output.

- **refinement w/ feedback**: self-refines the initial output based on a feedback. In the feedback, the LLM itself evaluates whether the output text complies with each instruction. This feedback and refinement process is repeated $T$ times.

- **refinement w/ feedback+each**: self-refines the initial output based on a feedback similar to "refinement w/ feedback", but this method gives feedback by resetting the prompt for each instruction. In other words, this method runs the LLM for the number of times as many as instructions to obtain feedbacks for each instruction.

- **refinement w/ feedback+each+cot (our method)**: See Section 5.1. Our method is regarded as self-refinement by self-feedback for each instructions using Zero-shot-CoT.

The performance comparison results are shown in Table 2 for GPT-4o and Claude 3.5 Sonnet. We confirmed that self-refinement and feedback-each are important contributors for the performance improvement. We also confirmed that the highest performance is achieved when all the components (self-refinement, self-feedback, Chain-of-Thought and feedback per instruction) are combined. Interestingly, just giving self-feedback does not improve the performance (See the performance change from "refinement w/o feedback" to "refinement w/ feedback"). This result indicates that the quality of self-feedback is important for the performance improvement. See Appendix D for the results of the other models.

## 5.3 FURTHER ANALYSIS

Based on the finding that the quality of self-feedback is important for the performance improvement, we further analyze what characteristics are important for the self-feedback. First, we conduct the following additional baseline experiments:

- **refinement w/ oracle**: uses an external oracle feedback mechanism that can precisely determine whether each instruction was successfully followed. Specifically, this method (1) generates an initial answer; (2) uses the oracle verifier to assess success or failure in following each instruction; (3) lists the instructions that were not followed and prompt the model to correct its answer accordingly. We repeat steps (2) and (3) for T times.

- **refinement w/ all false**: informs the model that it failed to follow all instructions. Specifically, this method (1) generates an initial answer; (2) prompt the model with all the instructions to inform the model that it failed to follow all the instructions and correct its answer accordingly. We repeats steps (2) for T times.

Table 2: Method comparison result on GPT-4o (Top) and Claude 3.5 Sonnet (Bottom). **Bold** means the best and underline means the second best score.

| [GPT-4o] | Prompt-level | Inst-level | Feedback Performance | | |
|---|---|---|---|---|---|
| Method | Accuracy | Accuracy | Precision | Recall | F1 Score |
| zero-shot | 0.484 | 0.861 | - | - | - |
| refinement w/o feedback | 0.560 | 0.887 | - | - | - |
| refinement w/ feedback | 0.558 | 0.887 | 0.79 | 0.44 | 0.56 |
| refinement w/ feedback+each | 0.615 | 0.908 | **0.84** | 0.16 | 0.27 |
| refinement w/ feedback+each+cot (**Ours**) | **0.621** | **0.909** | 0.82 | **0.76** | **0.79** |
| refinement w/ oracle | 0.700 | 0.929 | 1.0 | 1.0 | 1.0 |
| refinement w/ all false | 0.613 | 0.905 | 1.0 | 0 | 0 |

| [Claude 3.5] | Prompt-level | Inst-level | Feedback Performance | | |
|---|---|---|---|---|---|
| Method | Accuracy | Accuracy | Precision | Recall | F1 Score |
| zero-shot | 0.683 | 0.925 | - | - | - |
| refinement w/o feedback | 0.709 | 0.936 | - | - | - |
| refinement w/ feedback | 0.703 | 0.935 | 0.66 | **0.96** | 0.78 |
| refinement w/ feedback+each | 0.725 | 0.939 | 0.70 | 0.72 | 0.71 |
| refinement w/ feedback+each+cot (**Ours**) | **0.758** | **0.949** | **0.77** | 0.94 | **0.85** |
| refinement w/ oracle | 0.853 | 0.967 | 1.0 | 1.0 | 1.0 |
| refinement w/ all false | 0.792 | 0.955 | 1.0 | 0 | 0 |

The result is shown at the bottom sides of Table 2. When correct feedback is provided, answer correction leads to substantial improvements in instruction-following performance (refinement w/ oracle). Even when using "all false" feedback—informing the model that it failed to follow all instructions—performance improvement was observed, though not as significant as oracle feedback.

Based on the result, we make the following hypothesis: It is better to pick up all the instructions that are not being followed, even though the prediction includes mistakes, than to overlook the instructions that are not being followed. Here we define the confusion matrix of self-feedback prediction as follows. False-positive: mistakenly judging an instruction that is being followed as not being followed. False-negative: mistakenly judging an instruction that is not being followed as being followed. Based on this definition of the confusion matrix, we calculate precision, recall, and F1 score for each method. The result is shown at the right sides of Table 2. This result demonstrates that instruction-following performance has strong relationship with the precision: As the precision becomes higher, accuracy of the method tends to become high. This observation indicates that our hypothesis is correct. See Appendix E for the detail of the confusion matrix of some models.

## 6 CONCLUSION AND FUTURE WORK

We hava introduced ManyIFEval, a large-scale benchmark dataset comprising task prompts with up to ten objectively verifiable instructions. We conducted experiments based on ManyIFEval with various models, demonstrating that as the instruction count rises, the models' ability to follow individual instruction deteriorates gradually but constantly. As a result, the models' ability to follow all the instructions significantly drops by the "curse of instructions". To remove the curse without re-training models, we proposed a inference-time method that enhances performance through iterative self-refinement by feedback with instruction-level chain-of-thought reasoning. Our Experiments showed that our method drastically improved the performance of following multiple instructions.

It is important to note that the instructions used in our study were designed to be simple and allowed for objective, binary evaluations of success or failure in instruction following. These instructions did not encompass more abstract or nuanced directives, such as those found in legal texts, where judgments may have gradations and are less clear-cut. Addressing the challenges posed by such complex instructions remains an area for future work. Developing methods for LLMs to handle abstract instructions with subjective success criteria will be crucial for advancing their applicability in real-world scenarios that demand nuanced understanding and interpretation.

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

Appendix

## A    STATISTICS OF MANYIFEVAL.

Table 3 describes Statistics of ManyIFEval.

Table 3: A overview of ManyIFEval.

| #Samples | #Prompts | Avg. Len (word) |
|---|---|---|
| 1000 | 100 | 92.3 |

## B    INSTRUCTIONS IN MANYIFEVAL

Table 4 summarizes a list of Instructions in ManyIFEval.

Table 4: The list of 15 verifiable instructions, with brief descriptions from IFEval paper and Inst-level accuracy by gpt-4o-2024-05-13.

| Instruction Group | Instruction | Description | Inst-level acc |
|---|---|---|---|
| Keywords | Include Keywords | Include keywords {keyword1}, {keyword2} in your response. | 98% |
| Keywords | Keyword Frequency | In your response, the word should appear $N$ times. | 96% |
| Keywords | Forbidden Words | Do not include keywords (forbidden words) in the response. | 95% |
| Keywords | Letter Frequency | In your response, the letter {letter} should appear $N$ times. | 62% |
| Length Constraints | Number Paragraphs | Your response should contain $N$ paragraphs. You separate paragraphs using the markdown divider \\. | 87% |
| Length Constraints | Number Words | Answer with at least/around/at most $N$ words. | 81% |
| Length Constraints | Number Sentences | Answer with at least/around/at most $N$ sentences. | 73% |
| Detectable Content | Number Placeholder | The response must contain at least $N$ placeholders represented by square brackets, such as [address]. | 95% |
| Detectable Format | Number Bullets | Your answer must contain exactly $N$ bullet points. Use the markdown bullet points such as `- This is a point.` | 88% |
| Detectable Format | Title | Your answer must contain a title, wrapped in double quotation marks, such as "{title of your response}." | 100% |
| Change Cases | All Uppercase | Your entire response should be in English, and all capital letters only. | 97% |
| Change Cases | All Lowercase | Your entire response should be in English, and in all lowercase letters. | 97% |
| Change Cases | Frequency of all-capital words | In your response, words in all capital letters should appear at least/around/at most $N$ times. | 87% |
| Start with / End with | Quotation | Wrap your entire response with double quotation marks. | 100% |
| Punctuation | No Commas | In your entire response, refrain from the use of any commas. | 100% |

## C  EFFECT OF MULTIPLE UNIQUE INSTRUCTIONS ON PERFORMANCE

Previous studies have shown that increasing the input token length can negatively impact a model's reasoning capabilities (Levy et al., 2024; Liu et al., 2024). To determine whether the observed degradation was due to the increased number of instructions or simply the longer input sequences, we conducted an experiment where we repeated the same instructions to match the input length of prompts with multiple unique instructions. By measuring instruction-following performance with repeated instructions, we isolated the effect of input length from the effect of instruction variety. As shown in Figure 8, the degradation in performance with repeated instructions was less severe compared to when the number of unique instructions increased. In some cases, models even demonstrated improved instruction-following when instructions were repeated. These findings suggest that the negative impact on instruction-following performance is more attributable to the increased number of unique instructions rather than the overall input length, highlighting a limitation in the models' capacity to handle multiple instructions simultaneously.

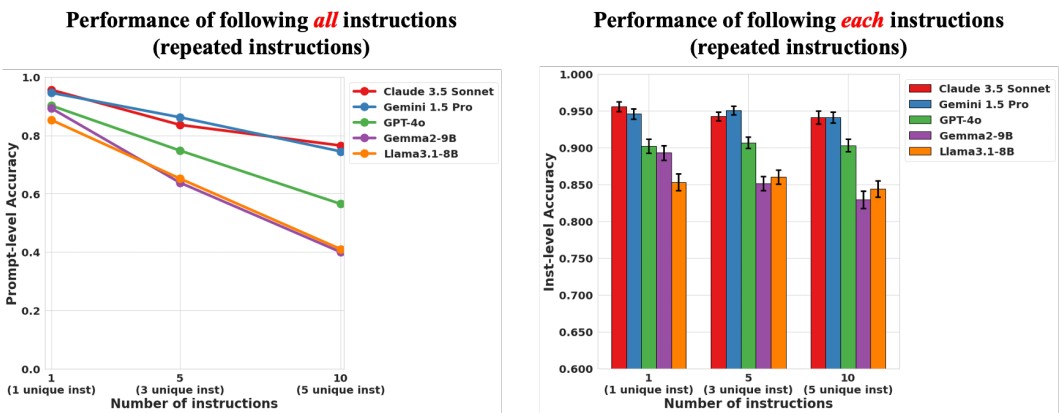

Figure 8: Multiple unique instructions affect performance.

## D  ADDITIONAL EXPERIMENT RESULTS ON METHOD COMPARISON

Table 5 describes additional experiment results on method comparison.

Table 5: Result of Gemini 1.5 Pro. **Bold** means the best and underline means the second best score.

| Method | Prompt-level Accuracy | Inst-level Accuracy | Feedback Performance | | |
|---|---|---|---|---|---|
| | | | Precision | Recall | F1 Score |
| zero-shot | 0.677 | 0.928 | - | - | - |
| refinement w/o feedback | 0.648 | 0.913 | - | - | - |
| refinement w/ feedback | 0.677 | 0.927 | 0.58 | 0.62 | 0.60 |
| refinement w/ feedback+each | **0.692** | **0.929** | **0.92** | 0.12 | 0.21 |
| refinement w/ feedback+each+cot (**Ours**) | 0.678 | 0.928 | 0.85 | **0.63** | **0.72** |
| refinement w/ oracle | 0.763 | 0.945 | 1.0 | 1.0 | 1.0 |
| refinement w/ all false | 0.730 | 0.940 | 1.0 | 0 | 0 |

## E  CONFUSION MATRIX OF ZERO-SHOT-COT-EACH.

Figure 9 describes a confusion matrix of zero-shot-cot-each method. The result shows that zero-shot-cot-each is a high precision approach.

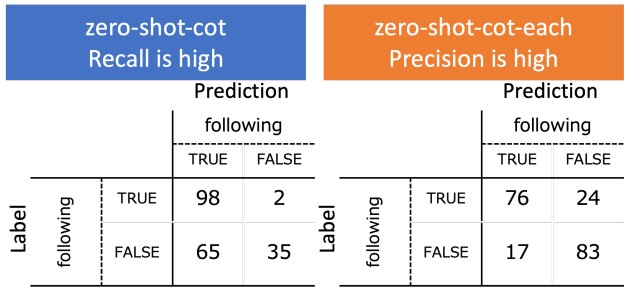

Figure 9: zero-shot-cot-each is high precision approach.

## F    THE DETAIL EXPERIMENT RESULT OF CURSE OF INSTRUCTIONS

Figure 10 describes the detail experiment result of curse of instructions.

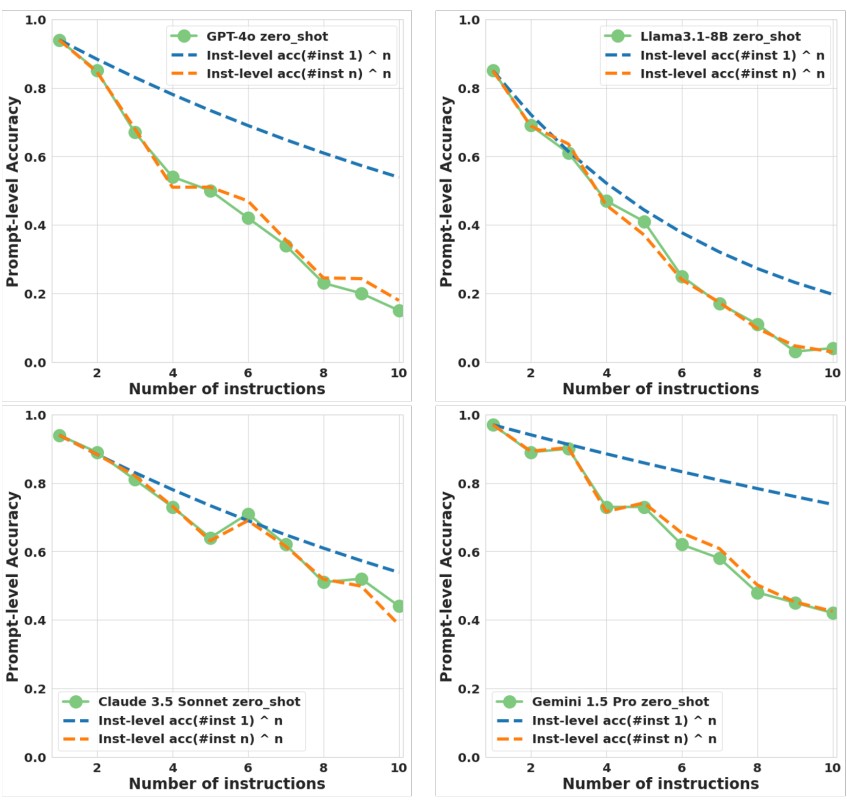

Figure 10: As the number of instructions increases, the models' ability to follow individual instruction deteriorates gradually but constantly. As a result, the models' ability to follow all the instructions significantly drops: the success rate of all the instructions is precisely explained by the success rate of individual instructions to the power of total number of instructions.

## G    EXAMPLE OF GENERATION

Table 6 and Table 7 describes Examples of prompts and LLMs' response for each method.

Table 6: Example response of GPT-4o zero-shot for 10 instructions. Red text indicates instructions that are not being followed.

| **Example response of GPT-4o zero-shot for 10 instructions** | |
|---|---|
| **Prompt** | Are the weather conditions in the Arctic very cold most of the year?

Your response should follow the instructions below:
- The response must contain at least 3 placeholders represented by square brackets, such as [address].
- Your answer must contain exactly 2 bullet points. Use the markdown bullet points such as: * This is point 1. * This is point 2
- Wrap your entire response with double quotation marks.
- In your response, the word fail should appear less than 2 times.
- In your response, the letter b should appear less than 8 times.
- Your answer must contain a title, wrapped in double angular brackets, such as <<poem of joy>>.
- In your response, words with all capital letters should appear at least 14 times.
- In your entire response, refrain from the use of any commas.
- There should be 2 paragraphs. Paragraphs are separated with the markdown divider: ***
- Do not include keywords ['no', 'yes'] in the response. |
| **Response** | "<<ARCTIC WEATHER CONDITIONS>>"

"The weather conditions in the Arctic are EXTREMELY COLD for most of the year. The region experiences long periods of DARKNESS during the winter months which contributes to the FRIGID temperatures. The average temperature in the Arctic can drop to as low as [temperature] degrees Fahrenheit during the winter.

***

* The summer months in the Arctic are SHORT and the temperatures rarely rise above [temperature] degrees Fahrenheit. Even during this time the weather can be quite CHILLY. The ARCTIC is known for its ICE and SNOW which cover the landscape for the majority of the year. The harsh conditions make it a CHALLENGING environment for both humans and wildlife." |

Table 7: Example response of GPT-4o feedback+cot+each for 10 instructions. All instructions are followed by feedback+each+cot.

| Example response of GPT-4o feedback+cot+each for 10 instructions | |
|---|---|
| **Prompt** | Are the weather conditions in the Arctic very cold most of the year? 

 Your response should follow the instructions below: 
 - The response must contain at least 3 placeholders represented by square brackets, such as [address]. 
 - Your answer must contain exactly 2 bullet points. Use the markdown bullet points such as: * This is point 1. * This is point 2 
 - Wrap your entire response with double quotation marks. 
 - In your response, the word fail should appear less than 2 times. 
 - In your response, the letter b should appear less than 8 times. 
 - Your answer must contain a title, wrapped in double angular brackets, such as <<poem of joy>>. 
 - In your response, words with all capital letters should appear at least 14 times. 
 - In your entire response, refrain from the use of any commas. 
 - There should be 2 paragraphs. Paragraphs are separated with the markdown divider: *** 
 - Do not include keywords ['no', 'yes'] in the response. |
| **Response** | "<<ARCTIC WEATHER CONDITIONS>>" 

 "The weather conditions in the Arctic are EXTREMELY COLD for most of the year. The region experiences LONG WINTERS with temperatures often dropping to EXTREMELY LOW levels. During the summer months the temperatures may rise slightly but they remain quite chilly compared to other regions like [location1]. The EXTREME COLD can be quite UNFORGIVING. 

 *** 
 * The Arctic is known for its HARSH CLIMATE which includes STRONG WINDS and HEAVY SNOWFALL. This makes [activity1] particularly challenging. The EXTREME COLD and STRONG WINDS can make it difficult to perform even simple tasks. 
 * Due to the EXTREME COLD and ISOLATED LOCATION the Arctic is a challenging environment for both humans and wildlife. The conditions can be quite UNFORGIVING and require SPECIALIZED EQUIPMENT and clothing to navigate safely. For instance [item1] is essential for survival in the EXTREME COLD. 

 Overall the Arctic's weather is characterized by its EXTREME COLD and LONG WINTERS making it a unique and challenging environment." |

