# OpenReview forum: "Curse of Instructions: Large Language Models Cannot Follow Multiple Instructions at Once"
_ICLR.cc/2025/Conference — Submitted to ICLR 2025_

### Official Review · Reviewer_k32n · 2024-10-20

**Soundness:** 2
**Presentation:** 3
**Contribution:** 2
**Rating:** 3
**Confidence:** 4

**Summary:**

The paper first introduces ManyIFEval, which is a dataset comprising task prompts with up to ten objectively verifiable instructions. Through this benchmark, the paper shows that current LLMs fails to comply to multiple instructions at once. To mitigate this problem, the paper apply a inference-time self-refinement strategy which boosts the performance.

**Strengths:**

- The paper is clearly written and straightforward.
- The reviewer agrees with the problem formulation; we need a LLM that can solve multiple instructions efficiently.

**Weaknesses:**

- In related works (Figure 2), the authors claim that the contribution of ManyIFEval lies on complexity. However, the complexity does not significantly differ from ComplexBench.
- The size of the benchmark is too small and limited. It is specified that there are 110 for training, 100 for testing, and 6 for few-shot prompting which is a small number of prompts for evaluation. Also, the paper evaluates on only 15 instructions which is very narrow and distinct from real use cases mentioned in Figure 1 (legal and medical).
- There are no further analysis depending on the model size or the  pretraining data scale for the result of Figure 1.
- In Section 4.3, the authors claim that the finding of the paper is a 'rule': however, the paper only investigates 5 LLMs with a limited evaluation dataset. The paper should provide correlation or similar quantitative measure to claim that the finding is a 'rule'.
- It is expected that the performance reduces as the number of instructions increases; humans would also struggle to generate a response correctly when provided with multiple instructions.
- The proposed approach in Section 5 shows comparable performance with a heuristic baseline 'refinement w/ all false'. This weakens the effectiveness of the proposed approach; even though the baseline is much simpler with similar inference costs, it performs similarly.
- Performance of another heuristic baseline is missing: conditioning on task prompt $P$, single instruction $I_{i}$, and previous output $O_{i-1}$ (repeating this for the number of instructions). (1st inference: $P$, $I_0$ -> $O_0$, 2nd inference $P$, $I_1$,  $O_0$,-> $O_1$,... )

**Questions:**

- For the current setup, the task prompt is fixed and there are multiple instructions for each instance. Would a similar finding be observed when there are multiple task prompts and corresponding instruction for each instance (ex) Task 1, Instruction 1, Task 2, Instruction 2, .., )
- Would few-shot ICL mitigate the curse of instructions?

---

> ### Author Response · Authors · 2024-11-23
> **Author response(1/4)**
>
> We thank the reviewer for the constructive feedback. Please let us know if our responses in the following address your concerns.
>
> ## About ManyIFEval contribution
> Responding to the following comments by the reviewer.
> > the complexity does not significantly differ from ComplexBench.
>
> > The size of the benchmark is too small and limited. It is specified that there are 110 for training, 100 for testing, and 6 for few-shot prompting which is a small number of prompts for evaluation.
>
> > Also, the paper evaluates on only 15 instructions which is very narrow and distinct from real use cases mentioned in Figure 1 (legal and medical).
>
> Our benchmark focuses on complexity in terms of the **number of instructions**, aiming to investigate the relationship between the number of instructions that must be satisfied simultaneously and the instruction following performance. In contrast, ComplexBench[1] is a benchmark that considers **nesting depth**—the hierarchical structure within a single instruction—as the measure of complexity. ComplexBench compares nesting depths only in the categories of 1, 2, and 3 or more. Moreover, the number of samples and the types of instructions provided are not consistent across different nesting depths in ComplexBench.
>
> In our benchmark, we have standardized both the number of samples and the types of instructions provided for each instruction count. We believe that it is precisely because we carefully designed the dataset in this manner that we were able to discover the "curse of instructions."
>
> At the time of our paper submission, our dataset contained 110 training samples and 100 test samples for each instruction count, totaling 1,100 training samples and 1,000 test samples. This was comparable to ComplexBench, which has the largest number of samples among existing instruction following benchmarks shown in Table 1 of our paper. However, considering the feedback that the limited number of samples might undermine reliability, **we have prepared a new dataset with ten times the number of samples—that is, 1,000 samples for each instruction count, totaling 10,000 samples.** The results for Gemini and GPT-4 are shown in the next response.
>
> **As with our initial submission, we confirmed that the performance of following all instructions does not maintain a high level until the number of instructions exceeds a certain threshold; instead, the performance transitions according to a power law.** We also observed that as the number of instructions increases, the instruction-level performance—that is, the ability to follow each individual instruction—deteriorates.
>
> As the reviewer pointed out, the instructions we handle are primarily related to character counts and formatting and are not limited to specialized domains such as law or medicine. In cases where the instructions pertain to domains like law or medicine, the instruction following performance for each instruction may be lower. Additionally, when there are many instructions that must be satisfied simultaneously, we believe that the influence of the "curse of instructions" would be even more pronounced. This is because, as confirmed in our recent experiments, the performance of following all instructions does not maintain a high level beyond a certain threshold number of instructions but instead transitions according to a power law.
>
> [1] Wen, B., Ke, P., Gu, X., Wu, L., Huang, H., Zhou, J., Li, W., Hu, B., Gao, W., Xu, J., & others. (2024). Benchmarking Complex Instruction-Following with Multiple Constraints Composition. ArXiv Preprint ArXiv:2407.03978.

---

> ### Author Response · Authors · 2024-11-23
> **Author response(2/4)**
>
> Result of gpt-4o-2024-05-13
>
> | # of instructions | Inst-level performance | Prompt-level performance (Simulation1) | Prompt-level performance (Simulation2) | Prompt-level performance (Experiments) | Inst 1st success | Inst 2nd success | Inst 3rd success | Inst 4th success | Inst 5th success | Inst 6th success | Inst 7th success | Inst 8th success | Inst 9th success | Inst 10th success |
> | --- | --- | --- | --- | --- | --- | --- | --- | --- | --- | --- | --- | --- | --- | --- |
> | 1  | 0.91 | 0.91 | 0.91 | 0.91 | 0.91 |     |     |     |     |     |     |     |     |     |
> | 2  | 0.90 | 0.83 | 0.81 | 0.82 | 0.92 | 0.88 |     |     |     |     |     |     |     |     |
> | 3  | 0.89 | 0.76 | 0.71 | 0.71 | 0.91 | 0.87 | 0.91 |     |     |     |     |     |     |     |
> | 4  | 0.88 | 0.69 | 0.60 | 0.60 | 0.90 | 0.86 | 0.89 | 0.89 |     |     |     |     |     |     |
> | 5  | 0.87 | 0.63 | 0.49 | 0.51 | 0.89 | 0.86 | 0.87 | 0.86 | 0.87 |     |     |     |     |     |
> | 6  | 0.87 | 0.57 | 0.43 | 0.45 | 0.88 | 0.86 | 0.88 | 0.87 | 0.86 | 0.90 |     |     |     |     |
> | 7  | 0.87 | 0.51 | 0.35 | 0.37 | 0.86 | 0.85 | 0.88 | 0.86 | 0.87 | 0.88 | 0.86 |     |     |     |
> | 8  | 0.87 | 0.46 | 0.30 | 0.33 | 0.86 | 0.86 | 0.89 | 0.86 | 0.86 | 0.88 | 0.86 | 0.89 |     |     |
> | 9  | 0.86 | 0.41 | 0.22 | 0.25 | 0.86 | 0.84 | 0.87 | 0.84 | 0.85 | 0.85 | 0.85 | 0.87 | 0.86 |     |
> | 10 | 0.85 | 0.37 | 0.18 | 0.19 | 0.86 | 0.84 | 0.87 | 0.84 | 0.85 | 0.86 | 0.85 | 0.88 | 0.85 | 0.84 |
>
> Result of gemini-1.5-pro-002
> | # of instructions | Inst-level performance | Prompt-level performance (Simulation1) | Prompt-level performance (Simulation2) | Prompt-level performance (Experiments) | Inst 1st success | Inst 2nd success | Inst 3rd success | Inst 4th success | Inst 5th success | Inst 6th success | Inst 7th success | Inst 8th success | Inst 9th success | Inst 10th success|
> | --- | --- | --- | --- | --- | --- | --- | --- | --- | --- | --- | --- | --- | --- | --- |
> | 1  | 0.95 | 0.95 | 0.95 | 0.95 | 0.95 |     |     |     |     |     |     |     |     |     |
> | 2  | 0.94 | 0.90 | 0.89 | 0.89 | 0.96 | 0.93 |     |     |     |     |     |     |     |     |
> | 3  | 0.94 | 0.85 | 0.83 | 0.83 | 0.95 | 0.93 | 0.94 |     |     |     |     |     |     |     |
> | 4  | 0.93 | 0.81 | 0.76 | 0.77 | 0.95 | 0.92 | 0.94 | 0.93 |     |     |     |     |     |     |
> | 5  | 0.93 | 0.77 | 0.68 | 0.70 | 0.95 | 0.92 | 0.94 | 0.93 | 0.92 |     |     |     |     |     |
> | 6  | 0.93 | 0.73 | 0.65 | 0.66 | 0.94 | 0.92 | 0.95 | 0.92 | 0.93 | 0.94 |     |     |     |     |
> | 7  | 0.93 | 0.68 | 0.59 | 0.60 | 0.94 | 0.92 | 0.94 | 0.93 | 0.91 | 0.94 | 0.93 |     |     |     |
> | 8  | 0.92 | 0.64 | 0.51 | 0.53 | 0.94 | 0.92 | 0.93 | 0.92 | 0.91 | 0.94 | 0.92 | 0.92 |     |     |
> | 9  | 0.92 | 0.60 | 0.44 | 0.47 | 0.93 | 0.91 | 0.93 | 0.92 | 0.91 | 0.93 | 0.93 | 0.92 | 0.91 |     |
> | 10 | 0.92 | 0.56 | 0.38 | 0.40 | 0.93 | 0.91 | 0.92 | 0.92 | 0.91 | 0.92 | 0.92 | 0.91 | 0.92 | 0.91 |

---

> ### Author Response · Authors · 2024-11-23
> **Author response(3/4)**
>
> ## About curse of instructions
> Responding to the following comments by the reviewer.
> > It is expected that the performance reduces as the number of instructions increases; humans would also struggle to generate a response correctly when provided with multiple instructions.
>
> > In Section 4.3, the authors claim that the finding of the paper is a 'rule': however, the paper only investigates 5 LLMs with a limited evaluation dataset. The paper should provide correlation or similar quantitative measure to claim that the finding is a 'rule'.
>
> > There are no further analysis depending on the model size or the pretraining data scale for the result of Figure 1.
>
>
> It is said that humans can hold 7±2 pieces of information in short-term memory[1]. If we apply this assumption to language models, one might expect that the performance of successfully following all instructions would remain high until the number of instructions exceeds a certain threshold, after which the performance would dramatically decline. However, our observed results differed from these expectations.
>
> Notably, **the performance in following all instructions does not remain high until the number of instructions exceeds a certain threshold; instead, the performance transitions according to a power law.** By preparing up to 10 instructions, ensuring an equal number of samples for each, and conducting reliable evaluations based on a programmatic approach, we observed the results and empirically confirmed the existence of the "curse of instructions."
>
> We have empirically confirmed curse of instructions across a total of five models, including both actively developed closed models and open models. As the reviewer have pointed out, it is necessary to examine the patterns that emerge when altering the model size or data size. However, the curse of instructions we discovered indicates that as the number of instructions the model is required to follow increases, its ability to comply with each instruction deteriorates according to a power law. Therefore, even though increasing the model size or data size may change the benchmark performance, we expect that this trend of degradation—characterized by a power law—will be common across different models.
>
>
> [1] Miller, G. A. (1956). The magical number seven, plus or minus two: Some limits on our capacity for processing information. Psychological review, 63(2), 81.

---

> ### Author Response · Authors · 2024-11-23
> **Author response(4/4)**
>
> ## About proposed method and baselines
>
> > Performance of another heuristic baseline is missing: conditioning on task prompt $P$, single instruction $I_i$, and previous output $O_{i-1}$ (repeating this for the number of instructions). (1st inference: $P, I_0 \rightarrow O_0$, 2nd inference $P, I_1, O_0 \rightarrow O_1, \dots$)
>
> When new instructions are provided and the response is revised, there is a possibility that the answer may not adhere to the instructions given previously, leading to an expected decrease in performance in following all instructions. Since it is necessary to comply with all instructions, we consider it appropriate as a baseline to provide all instructions within the prompt.
>
> > The proposed approach in Section 5 shows comparable performance with a heuristic baseline 'refinement w/ all false'. This weakens the effectiveness of the proposed approach; even though the baseline is much simpler with similar inference costs, it performs similarly.
>
> It is indeed true that "refinement with all false" showed performance improvements comparable to our proposed method. However, compared to the oracle case, the necessity of pursuing accurate feedback performance becomes evident. We have confirmed that our proposed method enhances feedback performance and improves overall performance through revisions. From a practical standpoint, especially when deploying in real systems, our method is more reliable from a human perspective than treating all feedback as false, and we believe this direction holds promise for future developments.
>
> > Would few-shot ICL mitigate the curse of instructions?
>
> We have not conducted experiments on few-shot In-Context Learning (ICL). While it is expected that the performance in following individual instructions would improve, we believe that the overall performance in adhering to all instructions would still deteriorate according to a power law.
>
> > For the current setup, the task prompt is fixed and there are multiple instructions for each instance. Would a similar finding be observed when there are multiple task prompts and corresponding instruction for each instance (ex) Task 1, Instruction 1, Task 2, Instruction 2, .., )
>
> Thank you for suggesting the problem setting where multiple task prompts exist and instructions are provided for each task prompt. It appears to be an interesting experiment. However, could you please provide examples of concrete real-world use cases where such a problem setting is observed? We would like to refer to them for our experiments.

---

> > ### Comment · Reviewer_k32n · 2024-11-25
> >
> > Thank you for your response and additional information.
> > However, since many of the concerns are unresolved (W3: analysis depending on the model size or the pretraining data scale, W4: provide correlation or similar quantitative measure to claim that the finding is a 'rule', W5: refinement w/ all false performance, W6: Performance of another heuristic baseline), I will keep my rating.

---

### Official Review · Reviewer_ApPj · 2024-10-30

**Soundness:** 2
**Presentation:** 3
**Contribution:** 2
**Rating:** 5
**Confidence:** 4

**Summary:**

This work studies the problem of following multiple instructions at once. To create a dataset for this study, this work extracts prompts and instructions from IFEval and produces a new dataset called ManyIFEval where each prompt has 10 instructions. The main empirical finding of this paper is that when there are n instructions, the prompt-level accuracy is equal to the instruction-level accuracy to the power of n. This work also proposes a feedback+refine-based method to improve multi-instruction following.

**Strengths:**

1. Analyzing the model's behavior on following multiple instructions at once is a simple yet insightful angle.
2. The feedback+refine framework provides nice improvement on multi-instruction following.

**Weaknesses:**

1. My biggest concern of this paper is that the discovered rule seems to be a direct consequence of the instruction-level errors being randomly distributed across different examples. This seems to be an artifact of the way how the dataset is designed. I would imagine the finding to be different if the dataset containing examples with different level of difficulties.
2. The size of the dataset (100) is relatively small. In practice, this may lead to unstable evaluation metrics.
3. Besides the concern in weakness 1, there are a number of additional important  questions left unanswered for this study. Specifically,
    1. When there are n instructions, does the instruction following performance of the 1st instruction differs from the instruction following performance of the last (nth) instruction?
    2. How does the performance of the methods proposed in Sec. 5 change w.r.t the number of total instructions?
    3. Can we discover any pattern on how Inst-level Accuracy (n) changes w.r.t n? Finding such patterns allow us to extrapolate performance prediction from smaller n to larger n.
4. No details of the Zero-shot-CoT are mentioned in the paper.
5. This is a minor point, but the definition of "Precision" and "Recall" in Table 2 is a bit counter-intuitive.

**Questions:**

1. LINE 249 only mentions a training set and a test set. In that case, does all model development in Sec. 5 done on the test set? Will that have a risk of overfitting?
2. There are several typos in this paper:
    * ManyIFE**VAL** at LINE 099 and 102.
	* **O**ur at LINE 154
	* I'm not sure what "refinement w/o zero-shot" means in LINE 458.

---

> ### Author Response · Authors · 2024-11-23
> **Author response(1/3)**
>
> We sincerely thank the reviewer for their thoughtful and constructive feedback. We greatly appreciate the reviewer's meticulous attention to detail, including pointing out the typos. The reviewer's careful review has been invaluable to us.
> We kindly ask that the reviewer let us know if our responses below adequately address your concerns.
>
> ## About curse of instructions
> Responding to the following comments by the reviewer.
>
> > My biggest concern of this paper is that the discovered rule seems to be a direct consequence of the instruction-level errors being randomly distributed across different examples. This seems to be an artifact of the way how the dataset is designed. I would imagine the finding to be different if the dataset containing examples with different level of difficulties.
>
> > When there are n instructions, does the instruction following performance of the 1st instruction differs from the instruction following performance of the last (nth) instruction?
>
> > Can we discover any pattern on how Inst-level Accuracy (n) changes w.r.t n? Finding such patterns allow us to extrapolate performance prediction from smaller n to larger n.
>
> It is said that humans can hold 7±2 pieces of information in short-term memory[1]. If we apply this assumption to language models, one might expect that the performance of successfully following all instructions would remain high until the number of instructions exceeds a certain threshold, after which the performance would dramatically decline. However, our observed results differed from these expectations.
>
> Notably, **the performance in following all instructions does not remain high until the number of instructions exceeds a certain threshold; instead, the performance transitions according to a power law.** By preparing up to 10 instructions, ensuring an equal number of samples for each, and conducting reliable evaluations based on a programmatic approach, we observed the results and empirically confirmed the existence of the "curse of instructions."
>
> As the reviewer has noted, if we can estimate an $\alpha$ such that the success rate of following each number of instructions deteriorates depending on the number of instructions—that is, expressing $success(x_i, n)$ as $success(x_i, 1) * \alpha ^ n$—it would be possible to predict the performance at a given number of instructions. In this study, we have confirmed a general trend that the performance of following all instructions transitions according to a power-law rule. We consider the estimation of α for more precise trend prediction to be a subject for future research.
>
> Even if instructions with varying levels of difficulty are included, we expect that it remains unchanged that the probability $P(X)$ of successfully following all given instructions can be estimated as $\text{success}(x_1, n) \times \text{success}(x_2, n) \times \dotsb \times \text{success}(x_n, n)$.
>
> The reviewer mentioned the necessity of examining the relationship between the position of instructions in the prompt and performance. However, we have confirmed that there is no large difference regarding the position and instruction-following performance (as shown in the table in following response: Inst 1st success vs. Inst 5th success vs. Inst 10th success). Could the reviewer please explain why the reviewer believes that it is necessary to consider the relationship between position and performance?
>
> [1] Miller, G. A. (1956). The magical number seven, plus or minus two: Some limits on our capacity for processing information. Psychological review, 63(2), 81.

---

> ### Author Response · Authors · 2024-11-23
> **Author response(2/3)**
>
> ## About ManyIFEval
> Responding to the following comments by the reviewer.
>
> > The size of the dataset (100) is relatively small. In practice, this may lead to unstable evaluation metrics.
>
>
> At the time of our paper submission, our dataset contained 110 training samples and 100 test samples for each instruction count, totaling 1,100 training samples and 1,000 test samples. This was comparable to ComplexBench, which has the largest number of samples among existing instruction following benchmarks shown in Table 1 of our paper. However, considering the feedback that the limited number of samples might undermine reliability, **we have prepared a new dataset with ten times the number of samples—that is, 1,000 samples for each instruction count, totaling 10,000 samples.** The results for Gemini and GPT-4 are shown in the next response.
>
> **As with our initial submission, we confirmed that the performance of following all instructions does not maintain a high level until the number of instructions exceeds a certain threshold; instead, the performance transitions according to a power law.** We also observed that as the number of instructions increases, the instruction-level performance—that is, the ability to follow each individual instruction—deteriorates.
>
>
> Result of gpt-4o-2024-05-13
>
> | # of instructions | Inst-level performance | Prompt-level performance (Simulation1) | Prompt-level performance (Simulation2) | Prompt-level performance (Experiments) | Inst 1st success | Inst 2nd success | Inst 3rd success | Inst 4th success | Inst 5th success | Inst 6th success | Inst 7th success | Inst 8th success | Inst 9th success | Inst 10th success |
> | --- | --- | --- | --- | --- | --- | --- | --- | --- | --- | --- | --- | --- | --- | --- |
> | 1  | 0.91 | 0.91 | 0.91 | 0.91 | 0.91 |     |     |     |     |     |     |     |     |     |
> | 2  | 0.90 | 0.83 | 0.81 | 0.82 | 0.92 | 0.88 |     |     |     |     |     |     |     |     |
> | 3  | 0.89 | 0.76 | 0.71 | 0.71 | 0.91 | 0.87 | 0.91 |     |     |     |     |     |     |     |
> | 4  | 0.88 | 0.69 | 0.60 | 0.60 | 0.90 | 0.86 | 0.89 | 0.89 |     |     |     |     |     |     |
> | 5  | 0.87 | 0.63 | 0.49 | 0.51 | 0.89 | 0.86 | 0.87 | 0.86 | 0.87 |     |     |     |     |     |
> | 6  | 0.87 | 0.57 | 0.43 | 0.45 | 0.88 | 0.86 | 0.88 | 0.87 | 0.86 | 0.90 |     |     |     |     |
> | 7  | 0.87 | 0.51 | 0.35 | 0.37 | 0.86 | 0.85 | 0.88 | 0.86 | 0.87 | 0.88 | 0.86 |     |     |     |
> | 8  | 0.87 | 0.46 | 0.30 | 0.33 | 0.86 | 0.86 | 0.89 | 0.86 | 0.86 | 0.88 | 0.86 | 0.89 |     |     |
> | 9  | 0.86 | 0.41 | 0.22 | 0.25 | 0.86 | 0.84 | 0.87 | 0.84 | 0.85 | 0.85 | 0.85 | 0.87 | 0.86 |     |
> | 10 | 0.85 | 0.37 | 0.18 | 0.19 | 0.86 | 0.84 | 0.87 | 0.84 | 0.85 | 0.86 | 0.85 | 0.88 | 0.85 | 0.84 |
>
> Result of gemini-1.5-pro-002
> | # of instructions | Inst-level performance | Prompt-level performance (Simulation1) | Prompt-level performance (Simulation2) | Prompt-level performance (Experiments) | Inst 1st success | Inst 2nd success | Inst 3rd success | Inst 4th success | Inst 5th success | Inst 6th success | Inst 7th success | Inst 8th success | Inst 9th success | Inst 10th success|
> | --- | --- | --- | --- | --- | --- | --- | --- | --- | --- | --- | --- | --- | --- | --- |
> | 1  | 0.95 | 0.95 | 0.95 | 0.95 | 0.95 |     |     |     |     |     |     |     |     |     |
> | 2  | 0.94 | 0.90 | 0.89 | 0.89 | 0.96 | 0.93 |     |     |     |     |     |     |     |     |
> | 3  | 0.94 | 0.85 | 0.83 | 0.83 | 0.95 | 0.93 | 0.94 |     |     |     |     |     |     |     |
> | 4  | 0.93 | 0.81 | 0.76 | 0.77 | 0.95 | 0.92 | 0.94 | 0.93 |     |     |     |     |     |     |
> | 5  | 0.93 | 0.77 | 0.68 | 0.70 | 0.95 | 0.92 | 0.94 | 0.93 | 0.92 |     |     |     |     |     |
> | 6  | 0.93 | 0.73 | 0.65 | 0.66 | 0.94 | 0.92 | 0.95 | 0.92 | 0.93 | 0.94 |     |     |     |     |
> | 7  | 0.93 | 0.68 | 0.59 | 0.60 | 0.94 | 0.92 | 0.94 | 0.93 | 0.91 | 0.94 | 0.93 |     |     |     |
> | 8  | 0.92 | 0.64 | 0.51 | 0.53 | 0.94 | 0.92 | 0.93 | 0.92 | 0.91 | 0.94 | 0.92 | 0.92 |     |     |
> | 9  | 0.92 | 0.60 | 0.44 | 0.47 | 0.93 | 0.91 | 0.93 | 0.92 | 0.91 | 0.93 | 0.93 | 0.92 | 0.91 |     |
> | 10 | 0.92 | 0.56 | 0.38 | 0.40 | 0.93 | 0.91 | 0.92 | 0.92 | 0.91 | 0.92 | 0.92 | 0.91 | 0.92 | 0.91 |

---

> > ### Author Response · Authors · 2024-11-23
> > **Author response(3/3)**
> >
> > ## About proposed method
> >
> > Responding to the following comments by the reviewer.
> > > How does the performance of the methods proposed in Sec. 5 change w.r.t the number of total instructions?
> >
> > > No details of the Zero-shot-CoT are mentioned in the paper.
> >
> > > LINE 249 only mentions a training set and a test set. In that case, does all model development in Sec. 5 done on the test set? Will that have a risk of overfitting?
> >
> > As shown in the table at bottom, performance improvements have been observed with the proposed method regardless of the number of instructions. **Particularly, the magnitude of the performance improvement is greater when the number of instructions is large.**
> >
> > In particularly challenging cases in this benchmark, which involves a total of 10 instructions, we confirmed that using the proposed method improved performance from 15% to 31% for GPT-4o, from 44% to 58% for Claude 3.5 Sonnet, and from 4% to 12% for Llama3.1-8B. We would like to draw your attention to the effectiveness of our proposed method.
> >
> > Our proposed method does not require training; in other words, it is a method that does not involve updating the model's weight parameters.
> >
> > The Zero-shot Chain-of-Thought (CoT)[1] prompt used to obtain feedback is as follows:
> >
> > ```Please assess whether the response follow a given instruction.
> > Walk through your reasoning step by step and determine if the response follow the instruction. At the conclusion of your assessment of the instruction, return `True` if the response meets the instruction; otherwise, return `False`.
> > ```
> >
> >
> > Furthermore, the prompt used when making corrections is as follows:
> >
> > ```
> > Your response does not satisfy the following instructions:
> > - {instruction description}
> > - {instruction description}
> > ...
> > - {instruction description}
> > Please refine your response to satisfy the instructions. Just output refined response.
> > ```
> >
> > **Since our method does not update the model's weight parameters and does not utilize specific samples within the prompts, we believe that the risk of overfitting is low.**
> >
> >
> >
> >
> > Result of gpt-4o-2024-05-13
> >
> > | # of instructions | Prompt-level performance (proposed) | Prompt-level performance (zero_shot) | Inst-level performance (proposed) | Inst-level performance (zero_shot) |
> > | --- | --- | --- | --- | --- |
> > | 1  | 0.95 | 0.94 | 0.95 | 0.94 |
> > | 2  | 0.91 | 0.85 | 0.96 | 0.92 |
> > | 3  | 0.76 | 0.67 | 0.92 | 0.88 |
> > | 4  | 0.74 | 0.54 | 0.93 | 0.84 |
> > | 5  | 0.62 | 0.50 | 0.91 | 0.87 |
> > | 6  | 0.51 | 0.42 | 0.89 | 0.88 |
> > | 7  | 0.59 | 0.35 | 0.92 | 0.86 |
> > | 8  | 0.42 | 0.23 | 0.90 | 0.84 |
> > | 9  | 0.40 | 0.20 | 0.90 | 0.85 |
> > | 10 | 0.30 | 0.15 | 0.90 | 0.84 |
> >
> > [1] Kojima, T., Gu, S. (Shane), Reid, M., Matsuo, Y., & Iwasawa, Y. (2022). Large Language Models are Zero-Shot Reasoners. In S. Koyejo, S. Mohamed, A. Agarwal, D. Belgrave, K. Cho, & A. Oh (Eds.), Advances in Neural Information Processing Systems (Vol. 35, pp. 22199–22213). Curran Associates, Inc. https://proceedings.neurips.cc/paper_files/paper/2022/file/8bb0d291acd4acf06ef112099c16f326-Paper-Conference.pdf

---

> > > ### Comment · Reviewer_ApPj · 2024-11-25
> > >
> > > Thank the authors for the detailed response. However, I still have the same concern as I wrote in the 1st point of the weaknesses. The discovered rule between prompt-level accuracy and instruction-level accuracy seems to be strongly correlated with how you construct the dataset and how difficult each instruction is. For example, in an extreme case where we are studying 2-step prompts, half of the instructions are super easy, and half of the instructions are super hard, it is obvious that the prompt-level accuracy may be heavily influenced by dataset construction and the exact difficulty of the instructions. As this is one of the main findings in this paper, I believe it's worth taking a closer look at when and why this rule holds.

---

> ### Author Response · Authors · 2024-11-25
>
> We thank the reviewer for the reply and for sharing important concerns.
>
> We apologize for the lack of explanation. At the time of the paper submission, we calculated Simulation1 and Simulation2 as follows:
>
> Simulation1: $\text{Prompt-level accuracy (n)} = (\text{Instruction-level accuracy (1)})^n$
>
> Simulation2: $\text{Prompt-level accuracy (n)} = (\text{Instruction-level accuracy (n)})^n$
>
>
> However, in light of your initial review comments, we conducted simulations considering the individual difficulty of each instruction as follows:
>
> Let $x_i$ be each instruction, $\text{success}(x_i, n)$ be the success probability of following $x_i$ when $n$ instructions are given simultaneously, $X(x_1, x_2, \dots, x_{n-1}, x_n)$ be the set of instructions included in the prompt, and $P(X)$ be the probability of successfully complying with all given instructions. We empirically confirmed that as $n$ increases, $\text{success}(x_i, n)$ becomes lower than $\text{success}(x_i, 1)$, and that $P(X)$ can be estimated as:
>
> Simulation1: $P(X) = \text{success}(x_1, 1) \times \text{success}(x_2, 1) \times \dots \times \text{success}(x_n, 1)$,
>
> Simulation2: $P(X) = \text{success}(x_1, n) \times \text{success}(x_2, n) \times \dots \times \text{success}(x_n, n)$.
>
> The results of this simulation, conducted with a sample size ten times larger, are shown in the table we shared in Author Response (2/3) under "Prompt-level performance (Simulation1)" and "Prompt-level performance (Simulation2)".
>
> As shown in Table 4 of Appendix B, the instructions in ManyIFEval vary in difficulty.
>
> What the "curse of instructions" suggests is that the performance of following all instructions can be estimated as the product of the success rates of following each instruction individually.
>
> Suppose there are four instructions in a prompt: two super hard instructions with a success rate of 0.5 each when given individually, and two super easy instructions with a success rate of 0.9 each when given individually. The probability of successfully following all instructions is not just 0.5 (the success rate of the super hard instructions), but is actually lower—estimated as $0.5 \times 0.5 \times 0.9 \times 0.9 = 0.2025$ (Simulation1). Also, we showed actual performance is lower than Simulation1 because instruction following success rates of each instruction get worse when there are multiple instructions.This illustrates what the "curse of instructions" implies.

---

> > ### Comment · Reviewer_ApPj · 2024-12-02
> >
> > Thank the authors for the further clarification on the additional experiments. I have increased my rating from 3 to 5. However, I will not be able to further increase my score as I feel these changes are substantial (as it's for one of the core claims) so this work will probably benefit from another round of review.

---

### Official Review · Reviewer_yjTk · 2024-11-01

**Soundness:** 3
**Presentation:** 3
**Contribution:** 2
**Rating:** 3
**Confidence:** 3

**Summary:**

This paper extends the IfEval dataset by combining instructions to demonstrate how LLMs fail to follow compound instructions. The task prompts used are free-form generation but the authors incrementally add instructions (upto 10) from one of 6 categories (keyword inclusion, length constraint, Case requirements, Punctuation, Start/End, Formatting - eg use of bullets).  Zero-shot evaluations on GPT4o, Gemini 1.5 Pro and Claude Sonnet 3.5 demonstrate that the performance deteriorates as the number of instructions increase. The paper also suggests inference time improvements -- specifically, providing feedback and CoT based refinement for each instruction that fails. Experiments have also been included where the feedback is returned by the oracle verifier. Interestingly, simply giving feedback that all instructions were a failure (regardless of whether that was true) results in a significant improvement of performance.

**Strengths:**

- Simple extension to IFEval
- Interesting observation of Instruction-following feedback (as referenced in summary)
- Programmatic evaluation

**Weaknesses:**

- This work inherits the weaknesses of IFEval -- for instance, task level performance is never assessed
- refinement w/ feedback+each+cot appears to be a very expensive solution (no discussion in paper)
- Evaluation only on three (closed) models

**Questions:**

No questions

---

> ### Author Response · Authors · 2024-11-22
> **Author response(1/1)**
>
> We thank the reviewer for the constructive feedback. Please let us know if our responses in the following address your concerns.
> > This work inherits the weaknesses of IFEval -- for instance, task level performance is never assessed
>
> In our benchmark, we measure performance based on whether each objectively verifiable instruction was followed and whether all instructions were adhered to.
> Could you tell us what task-level performance means?
> Additionally, what other weaknesses of IFEval do you consider significant? We would like to address these as part of future work, so we would greatly appreciate your insights.
>
> > refinement w/ feedback+each+cot appears to be a very expensive solution (no discussion in paper)
>
> It is true that this method incurs higher computational costs during inference compared to zero-shot inference. However, it has been reported that improving performance by increasing computational costs during inference is more cost-effective than investing heavily in training costs. This is a well-researched topic due to its demonstrated efficacy [1, 2]. As a future research, We aim to address the task of fine-tuning to improve performance, verifying the extent of performance gains relative to the additional computational costs incurred, and comparing these results with the proposed method.
>
> > Evaluation only on three (closed) models
>
> As mentioned in the paper, evaluations were conducted using open models such as Llama3 and Gemma2. Could you tell us what other models would be appropriate for evaluation?
>
> [1] Kumar, A., Zhuang, V., Agarwal, R., Su, Y., Co-Reyes, J. D., Singh, A., Baumli, K., Iqbal, S., Bishop, C., Roelofs, R., Zhang, L. M., McKinney, K., Shrivastava, D., Paduraru, C., Tucker, G., Precup, D., Behbahani, F., & Faust, A. (2024). Training Language Models to Self-Correct via Reinforcement Learning. https://arxiv.org/abs/2409.12917
>
> [2] Snell, C., Lee, J., Xu, K., & Kumar, A. (2024). Scaling LLM Test-Time Compute Optimally can be More Effective than Scaling Model Parameters. https://arxiv.org/abs/2408.03314

---

> > ### Comment · Reviewer_yjTk · 2024-11-26
> > **Acknowledgement**
> >
> > Thank you for the response.
> >
> > Task performance: If an instruction for writing an article on the sun says the text should be capitalized, checking only for capitalization does not say anything about whether the article was indeed about the sun (while being capitalized).
> >
> > I would like to retain my scores (I have read also read the other reviews and author responses).

---

### Official Review · Reviewer_aUoj · 2024-11-10

**Soundness:** 2
**Presentation:** 2
**Contribution:** 2
**Rating:** 3
**Confidence:** 4

**Summary:**

This paper proposed ManyIFEval, a large-scale benchmark dataset comprising task prompts with up to ten objectively verifiable instructions to test LLMs' ability of following multiple instructions. The authors conducted comprehensive analysis on ManyIFEVAL with different models including GPT-4o, Claude-3.5, Gemini-1.5, etc. The results suggested that models struggle to follow multiple instructions at once when scaling the number of instructions. The author also proposed a method to mitigate the performance degradation by iterative self-refinement through self-feedback loops in combination with chain-of-thought reasoning for each instruction.

**Strengths:**

- This paper is technically sound and the topic (LLMs' capability to follow multiple instructions at the same time) studied in the paper is valuable.
- The experiments are well designed and contains comprehensive analysis.

**Weaknesses:**

- The major concern I have regarding this paper is that the lack of excitement compared to the literature. First, as pointed out by the authors, the benchmark is an extension of IFEval (for both prompt construction and evaluation framework). The main ingradient added in this paper is to extend the number of instructions and balance the number of instructions per sample. Both are too trivial to establish a new benchmark in my opinion. Second, one of the major conclusions in the paper that "as instruction count rises, the models’ ability to follow individual instruction deteriorates gradually but constantly" is not new and it's already discussed in the ComplexBench. Third, the mitigation method (self-refinement) is also a widely adapted method to improve LLMs performance across different tasks. I am not sure if there are new insights community can gain from the mitigation on this specific benchmark.
- Section 4.3 and Figure 4 are quite confusing to me. To me, isn't that quite intuitive that if instruction-level accuracy is $p$, it means for each instruction, the probability of correct instruction following is $p$ on average. Then, for $n$ instructions, prompt-level accuracy is $p^n$, which is equation (3). Why do we need a simulation in Figure 4? Also, why is this a secret rule?

**Questions:**

- In Table 1, why "Model & Program" is marked as "X"? Does it suggest model-based eval is bad?
- In line 419, the repetition is set to 5. I am wondering have you tried different number of $T$ to see the trend of performance?
- In Table 2, do you have the performance numbers on instruction-level? I think put that number would also help better understand the results.

---

> ### Author Response · Authors · 2024-11-23
> **Author response(1/3)**
>
> We thank the reviewer for the constructive feedback. Please let us know if our responses in the following address your concerns.
>
> ## About ManyIFEval contribution
> Responding to the following comments by the reviewer.
> > The main ingredient added in this paper is to extend the number of instructions and balance the number of instructions per sample. Both are too trivial to establish a new benchmark in my opinion.
>
> > "as instruction count rises, the models’ ability to follow individual instruction deteriorates gradually but constantly" is not new and it's already discussed in the ComplexBench
>
> > In Table 1, why "Model & Program" is marked as "X"? Does it suggest model-based eval is bad?
>
> The purpose of the benchmark is to investigate the relationship between the number of instructions that must be satisfied simultaneously and the instruction following performance. Therefore, we considered it important to equalize the number of samples for each instruction count, ensure that there is not significant variance in the difficulty of following each instruction, and that the success or failure of instruction following can be reliably determined.
>
> As the reviewer pointed out, when comparing cases where the Nesting Depth is 1, 2, and 3 or more, it appears that the instruction-following performance decreases as the Depth increases (ComplexBench[1] Table 5). However, the evaluated Nesting Depths are only compared categorically as 1, 2, or 3 and above. Additionally, the number of samples and the types of instructions provided are not consistent for each Nesting Depth.
>
> In our benchmark, we standardized the number of samples and the types of instructions given for each instruction count. We believe that it was precisely because we designed the dataset so carefully that we were able to discover the "curse of instructions."
> Furthermore, while ComplexBench uses LLMs to judge instruction following, issues have been identified regarding evaluations using models. As can be seen in Table 4 of ComplexBench, the agreement rate with human judgment is about 87%.
>
> To ensure a highly reliable evaluation, we adopted an objective judgment method using programs. And that is why "Model & Program" is marked as "X".
>
> We believe that designing a benchmark aimed at systematically investigating the relationship between the number of instructions and instruction-following performance is a sufficient contribution to the community.
>
>
> [1] Wen, B., Ke, P., Gu, X., Wu, L., Huang, H., Zhou, J., Li, W., Hu, B., Gao, W., Xu, J., & others. (2024). Benchmarking Complex Instruction-Following with Multiple Constraints Composition. ArXiv Preprint ArXiv:2407.03978.
>
> [2] Wang, P., Li, L., Chen, L., Cai, Z., Zhu, D., Lin, B., Cao, Y., Kong, L., Liu, Q., Liu, T., & Sui, Z. (2024). Large Language Models are not Fair Evaluators. In L.-W. Ku, A. Martins, & V. Srikumar (Eds.), Proceedings of the 62nd Annual Meeting of the Association for Computational Linguistics (Volume 1: Long Papers) (pp. 9440–9450). Association for Computational Linguistics. https://doi.org/10.18653/v1/2024.acl-long.511
>
> [3] Shen, C., Cheng, L., Nguyen, X.-P., You, Y., & Bing, L. (2023). Large Language Models are Not Yet Human-Level Evaluators for Abstractive Summarization. In H. Bouamor, J. Pino, & K. Bali (Eds.), Findings of the Association for Computational Linguistics: EMNLP 2023 (pp. 4215–4233). Association for Computational Linguistics. https://doi.org/10.18653/v1/2023.findings-emnlp.278
>
> [4] Zheng, L., Chiang, W.-L., Sheng, Y., Zhuang, S., Wu, Z., Zhuang, Y., Lin, Z., Li, Z., Li, D., Xing, E., Zhang, H., Gonzalez, J. E., & Stoica, I. (2023). Judging LLM-as-a-Judge with MT-Bench and Chatbot Arena. Thirty-Seventh Conference on Neural Information Processing Systems Datasets and Benchmarks Track. https://openreview.net/forum?id=uccHPGDlao

---

> > ### Comment · Reviewer_aUoj · 2024-11-27
> > **Thanks for the response**
> >
> > Thank you for the response! However, I don't think it resolves my major concern about the excitement compared to what already exists in the literature. I would keep my rating unchanged.

---

> ### Author Response · Authors · 2024-11-23
> **Author response(2/3)**
>
> ## About curse of instructions
> Responding to the following comments by the reviewer.
> > isn't that quite intuitive that if instruction-level accuracy is , it means for each instruction, the probability of correct instruction following is  on average. Then, for  instructions, prompt-level accuracy is , which is equation (3). Why do we need a simulation in Figure 4? Also, why is this a secret rule?
>
> It is said that humans can hold 7±2 pieces of information in short-term memory[1]. If we apply this assumption to language models, one might expect that the performance of successfully following all instructions would remain high until the number of instructions exceeds a certain threshold, after which the performance would dramatically decline. However, our observed results differed from these expectations.
>
> Notably, the performance in following all instructions does not remain high until the number of instructions exceeds a certain threshold; instead, the performance transitions according to a power law. By preparing up to 10 instructions, ensuring an equal number of samples for each, and conducting reliable evaluations based on a programmatic approach, we observed the results and empirically confirmed the existence of the "curse of instructions."
>
> We believe that the "curse of instructions" is an issue that should be widely recognized, especially in use cases where multiple instructions need to be satisfied simultaneously. This problem becomes particularly pronounced when the individual instruction-following performance is low or when there are many instructions to follow.
>
> [1] Miller, G. A. (1956). The magical number seven, plus or minus two: Some limits on our capacity for processing information. Psychological review, 63(2), 81.

---

> > ### Comment · Reviewer_aUoj · 2024-11-27
> > **My concern still remains**
> >
> > Thanks again for the response!
> >
> > > one might expect that the performance of successfully following all instructions would remain high until the number of instructions exceeds a certain threshold
> >
> > Previous literature already shows that the instruction-following capability decrease as the number of instructions increase. Also, as I stated in my original review "it means for each instruction, the probability of correct instruction following is $p$ on average. Then, for instructions, prompt-level accuracy is $p^n$, which is equation (3)".

---

> ### Author Response · Authors · 2024-11-23
> **Author response(3/3)**
>
> ## About proposed method
> Responding to the following comments by the reviewer.
> > the mitigation method (self-refinement) is also a widely adapted method to improve LLMs performance across different tasks. I am not sure if there are new insights community can gain from the mitigation on this specific benchmark.
>
> > In line 419, the repetition is set to 5. I am wondering have you tried different number of
>  to see the trend of performance?
>
> Previous studies have pointed out that it is a challenging task for LLMs themselves to distinguish between success and failure in following instructions[1]. In our proposed method, correctly obtaining feedback on the success or failure of instruction following is a crucial factor. By separating multiple instructions and having the model judge compliance for each instruction individually, we have confirmed an improvement in feedback performance. Our method aligns with the findings[2], which observed that when human evaluators assess compliance for each instruction individually, the overall variance in the evaluation of instruction compliance decreases and reliability increases.
>
> As the reviewer have pointed out, the method of using feedback to revise answers has already been proposed. However, there is debate about which tasks such methods are effective for [3]. Confirming its effectiveness in the instruction-following task is one of our contributions.
>
> In the self-refine paper[4], three refinements were performed, and although the performance gains diminished with each additional correction, it was reported that increasing the number of corrections led to performance improvements. Considering that our task is more challenging than those addressed in self-refine, we conducted experiments with five corrections and did not experiment with other numbers of corrections. The performance gains per number of self-corrections were as follows.
>
> Result of gpt-4o-2024-05-13
>
> | Refinement Stage | Prompt-Level Performance | Inst-Level Performance |
> |----------------------------|--------------------------|-------------------------|
> | Zero-Shot (No Refinement) | 0.502 | 0.878 |
> | After 1 Refinement | 0.591 | 0.910 |
> | After 2 Refinements | 0.608 | 0.917 |
> | After 3 Refinements | 0.620 | 0.920 |
> | After 4 Refinements | 0.620 | 0.920 |
> | After 5 Refinements | 0.621 | 0.921 |
>
>
> Existing studies have confirmed performance improvements even when corrections were made up to a scale of $2^5$ iterations in mathematical tasks[5, 6].
>
> > In Table 2, do you have the performance numbers on instruction-level? I think put that number would also help better understand the results.
>
> Thank you for your suggestion. We have revised Table 2 and added the results for Inst-level. We have confirmed that the proposed method improves Inst-level performance.
>
> [1] Zeng, Z., Yu, J., Gao, T., Meng, Y., Goyal, T., & Chen, D. (2024). Evaluating Large Language Models at Evaluating Instruction Following. The Twelfth International Conference on Learning Representations. https://openreview.net/forum?id=tr0KidwPLc
>
> [2] Qin, Y., Song, K., Hu, Y., Yao, W., Cho, S., Wang, X., Wu, X., Liu, F., Liu, P., & Yu, D. (2024). InFoBench: Evaluating Instruction Following Ability in Large Language Models. In L.-W. Ku, A. Martins, & V. Srikumar (Eds.), Findings of the Association for Computational Linguistics ACL 2024 (pp. 13025–13048). Association for Computational Linguistics. https://aclanthology.org/2024.findings-acl.772
>
> [3] Kamoi, R., Zhang, Y., Zhang, N., Han, J., & Zhang, R. (2024). When Can LLMs Actually Correct Their Own Mistakes? A Critical Survey of Self-Correction of LLMs. Transactions of the Association for Computational Linguistics, 12, 1417–1440. https://doi.org/10.1162/tacl_a_00713
>
> [4] Madaan, A., Tandon, N., Gupta, P., Hallinan, S., Gao, L., Wiegreffe, S., Alon, U., Dziri, N., Prabhumoye, S., Yang, Y., Gupta, S., Majumder, B. P., Hermann, K., Welleck, S., Yazdanbakhsh, A., & Clark, P. (2023). Self-Refine: Iterative Refinement with Self-Feedback. In A. Oh, T. Naumann, A. Globerson, K. Saenko, M. Hardt, & S. Levine (Eds.), Advances in Neural Information Processing Systems (Vol. 36, pp. 46534–46594). Curran Associates, Inc. https://proceedings.neurips.cc/paper_files/paper/2023/file/91edff07232fb1b55a505a9e9f6c0ff3-Paper-Conference.pdf
>
> [5] Kumar, A., Zhuang, V., Agarwal, R., Su, Y., Co-Reyes, J. D., Singh, A., Baumli, K., Iqbal, S., Bishop, C., Roelofs, R., Zhang, L. M., McKinney, K., Shrivastava, D., Paduraru, C., Tucker, G., Precup, D., Behbahani, F., & Faust, A. (2024). Training Language Models to Self-Correct via Reinforcement Learning. https://arxiv.org/abs/2409.12917
>
> [6] Snell, C., Lee, J., Xu, K., & Kumar, A. (2024). Scaling LLM Test-Time Compute Optimally can be More Effective than Scaling Model Parameters. https://arxiv.org/abs/2408.03314

---

### Author Response · Authors · 2024-11-22
**Response to all reviewers (1/3)**

Thank you very much for your courteous and constructive feedback.
We have summarized our responses below to address the common points raised in your reviews.

## Curse of Instructions

Our assertion in this research is the discovery of the **"curse of instructions,"** where increasing the number of instructions leads to a deterioration in the instruction following performance of each instruction, and a dramatic decline in the performance of following all given instructions. **We empirically confirmed that the degree of performance degradation is influenced exponentially with respect to the number of instructions, demonstrating an important issue in use cases where multiple instructions need to be satisfied simultaneously.**

It is said that humans can hold 7±2 pieces of information in short-term memory[1]. If we apply this assumption to language models, one might expect that the performance of successfully following all instructions would remain high until the number of instructions exceeds a certain threshold, after which the performance would dramatically decline. However, our observed results differed from these expectations.

Regarding the relationship between the number of instructions and the performance of complying with all multiple instructions, we confirmed in **Simulation 1** that the trend in performance corresponds to the product of the performances when each instruction is given alone. By considering that the compliance performance for each instruction deteriorates as the number of simultaneously given instructions increases, we obtained simulation results (**Simulation 2**) that are closer to the experimental results.

Let $x_i$ be each instruction, $\text{success}(x_i, n)$ be the success probability of following  $x_i$ when $n$ instructions are given simultaneously, $X(x_1, x_2, \dots, x_{n-1}, x_n)$ be the set of instructions included in the prompt, and $P(X)$ be the probability of successfully complying with all given instructions. We empirically confirmed that as $n$ increases, $\text{success}(x_i, n)$ becomes lower than $\text{success}(x_i, 1)$, and that $P(X)$ can be estimated as:

$P(X) = \text{success}(x_1, n) \times \text{success}(x_2, n) \times \dots \times \text{success}(x_n, n)$

Notably, **the performance in following all instructions does not remain high until the number of instructions exceeds a certain threshold; instead, the performance transitions according to a power law.**

We believe that the significant deterioration in performance of following all instructions is a problem that should be widely recognized, especially in use cases where multiple instructions must be satisfied simultaneously. **This issue becomes particularly pronounced when the compliance performance for individual instructions is low or when there are many instructions to follow.**

[1] Miller, G. A. (1956). The magical number seven, plus or minus two: Some limits on our capacity for processing information. Psychological review, 63(2), 81.

---

> ### Author Response · Authors · 2024-11-22
> **Response to all reviewers (2/3)**
>
> ## ManyIFEval
>
> The purpose of the benchmark is to investigate the relationship between the number of instructions that must be satisfied simultaneously and the instruction following performance. Therefore, we considered it important to equalize the number of samples for each instruction count, ensure that there is not significant variance in the difficulty of following each instruction, and that the success or failure of instruction following can be reliably determined.
>
> By extracting a set of instructions from IFEval—where the success or failure of instruction following can be reliably judged based on program—that can be followed with high performance when given individually, and by equalizing the number of samples, we believe we have made a sufficient contribution toward achieving the purpose of the benchmark.
>
> In the paper, the number of samples was comparable to ComplexBench, which has the largest number of samples among existing instruction following benchmarks shown in Table 1. However, taking into account the comment that the sample size is small and therefore lacks reliability, **we have prepared new data with ten times the number of samples—that is, 1,000 samples for each instruction count, totaling 10,000 samples.** The results for Gemini and GPT-4 are shown in table in this response.
>
> From the trends in instruction following performance with respect to Inst-level accuracy and the order of instructions in the prompt, we confirmed that as the number of instructions increases, the performance of following each instruction deteriorates.
>
> Furthermore, by comparing the actual observed results with Simulation 1 ($\text{success}(x_i, 1)^n$) and Simulation 2 ($\text{success}(x_1, n)^n$), we empirically confirmed that the probability $P(X)$ of successfully following all the given instructions can be estimated as $\text{success}(x_1, n) \times \text{success}(x_2, n) \times \dots \times \text{success}(x_n, n)$.
>
> Result of gpt-4o-2024-05-13
> | # of instructions | Inst-level performance | Prompt-level performance (Simulation1) | Prompt-level performance (Simulation2) | Prompt-level performance (Experiments) | Inst 1st success | Inst 2nd success | Inst 3rd success | Inst 4th success | Inst 5th success | Inst 6th success | Inst 7th success | Inst 8th success | Inst 9th success | Inst 10th success |
> | --- | --- | --- | --- | --- | --- | --- | --- | --- | --- | --- | --- | --- | --- | --- |
> | 1  | 0.91 | 0.91 | 0.91 | 0.91 | 0.91 |     |     |     |     |     |     |     |     |     |
> | 2  | 0.90 | 0.83 | 0.81 | 0.82 | 0.92 | 0.88 |     |     |     |     |     |     |     |     |
> | 3  | 0.89 | 0.76 | 0.71 | 0.71 | 0.91 | 0.87 | 0.91 |     |     |     |     |     |     |     |
> | 4  | 0.88 | 0.69 | 0.60 | 0.60 | 0.90 | 0.86 | 0.89 | 0.89 |     |     |     |     |     |     |
> | 5  | 0.87 | 0.63 | 0.49 | 0.51 | 0.89 | 0.86 | 0.87 | 0.86 | 0.87 |     |     |     |     |     |
> | 6  | 0.87 | 0.57 | 0.43 | 0.45 | 0.88 | 0.86 | 0.88 | 0.87 | 0.86 | 0.90 |     |     |     |     |
> | 7  | 0.87 | 0.51 | 0.35 | 0.37 | 0.86 | 0.85 | 0.88 | 0.86 | 0.87 | 0.88 | 0.86 |     |     |     |
> | 8  | 0.87 | 0.46 | 0.30 | 0.33 | 0.86 | 0.86 | 0.89 | 0.86 | 0.86 | 0.88 | 0.86 | 0.89 |     |     |
> | 9  | 0.86 | 0.41 | 0.22 | 0.25 | 0.86 | 0.84 | 0.87 | 0.84 | 0.85 | 0.85 | 0.85 | 0.87 | 0.86 |     |
> | 10 | 0.85 | 0.37 | 0.18 | 0.19 | 0.86 | 0.84 | 0.87 | 0.84 | 0.85 | 0.86 | 0.85 | 0.88 | 0.85 | 0.84 |
>
> Result of gemini-1.5-pro-002
> | # of instructions | Inst-level performance | Prompt-level performance (Simulation1) | Prompt-level performance (Simulation2) | Prompt-level performance (Experiments) | Inst 1st success | Inst 2nd success | Inst 3rd success | Inst 4th success | Inst 5th success | Inst 6th success | Inst 7th success | Inst 8th success | Inst 9th success | Inst 10th success|
> | --- | --- | --- | --- | --- | --- | --- | --- | --- | --- | --- | --- | --- | --- | --- |
> | 1  | 0.95 | 0.95 | 0.95 | 0.95 | 0.95 |     |     |     |     |     |     |     |     |     |
> | 2  | 0.94 | 0.90 | 0.89 | 0.89 | 0.96 | 0.93 |     |     |     |     |     |     |     |     |
> | 3  | 0.94 | 0.85 | 0.83 | 0.83 | 0.95 | 0.93 | 0.94 |     |     |     |     |     |     |     |
> | 4  | 0.93 | 0.81 | 0.76 | 0.77 | 0.95 | 0.92 | 0.94 | 0.93 |     |     |     |     |     |     |
> | 5  | 0.93 | 0.77 | 0.68 | 0.70 | 0.95 | 0.92 | 0.94 | 0.93 | 0.92 |     |     |     |     |     |
> | 6  | 0.93 | 0.73 | 0.65 | 0.66 | 0.94 | 0.92 | 0.95 | 0.92 | 0.93 | 0.94 |     |     |     |     |
> | 7  | 0.93 | 0.68 | 0.59 | 0.60 | 0.94 | 0.92 | 0.94 | 0.93 | 0.91 | 0.94 | 0.93 |     |     |     |
> | 8  | 0.92 | 0.64 | 0.51 | 0.53 | 0.94 | 0.92 | 0.93 | 0.92 | 0.91 | 0.94 | 0.92 | 0.92 |     |     |
> | 9  | 0.92 | 0.60 | 0.44 | 0.47 | 0.93 | 0.91 | 0.93 | 0.92 | 0.91 | 0.93 | 0.93 | 0.92 | 0.91 |     |
> | 10 | 0.92 | 0.56 | 0.38 | 0.40 | 0.93 | 0.91 | 0.92 | 0.92 | 0.91 | 0.92 | 0.92 | 0.91 | 0.92 | 0.91 |

---

> ### Author Response · Authors · 2024-11-22
> **Response to all reviewers (3/3)**
>
> ## About the Proposed Method
> In particularly challenging cases in this benchmark, which involves a total of 10 instructions, we confirmed that using the proposed method improved performance from 15% to 31% for GPT-4o, from 44% to 58% for Claude 3.5 Sonnet, and from 4% to 12% for Llama3.1-8B. We would like to draw your attention to the effectiveness of our proposed method.
> As you have pointed out, the method of using feedback to revise answers has already been proposed. However, there is debate about which tasks such methods are effective for [1]. Confirming its effectiveness in the instruction-following task is one of our contributions.
>
> In developing our method, we first confirmed that if correct feedback can be obtained regarding instruction following, significant performance improvement can be expected through answer refinement (refinement w/ oracle in Table 2). Therefore, we considered it important to obtain accurate feedback on instruction adherence. We observed that when the model is asked to judge whether multiple instructions are followed, the feedback performance was not good (refinement w/ feedback in Table 2). By having the model assess adherence to each individual instruction, we confirmed that the feedback performance on its own answers improved (refinement w/ feedback+each+cot in Table 2).
> We believe this is an effective method that aligns with existing research [2], which suggests that in the evaluation of instruction following, dividing instructions and evaluating them individually leads to higher agreement among human evaluators and allows for more reliable judgments.
>
> While we confirmed performance improvements with the proposed method, we acknowledge your point regarding computational cost due to the increased number of inferences. Comparing the performance improvement margin with methods that are considered to have higher computational costs, such as fine-tuning, will be addressed in future work.
>
> [1] Kamoi, R., Zhang, Y., Zhang, N., Han, J., & Zhang, R. (2024). When Can LLMs Actually Correct Their Own Mistakes? A Critical Survey of Self-Correction of LLMs. Transactions of the Association for Computational Linguistics, 12, 1417–1440. https://doi.org/10.1162/tacl_a_00713
>
> [2] Qin, Y., Song, K., Hu, Y., Yao, W., Cho, S., Wang, X., Wu, X., Liu, F., Liu, P., & Yu, D. (2024). InFoBench: Evaluating Instruction Following Ability in Large Language Models. In L.-W. Ku, A. Martins, & V. Srikumar (Eds.), Findings of the Association for Computational Linguistics ACL 2024 (pp. 13025–13048). Association for Computational Linguistics. https://aclanthology.org/2024.findings-acl.772

---

### Meta-Review · Area_Chair_nFPR · 2024-12-21

**Metareview:**

The submission proposed one question for LLMs: the challenges in following multiple instructions at the same time. The authors introduce a benchmark, ManyIFEval, with prompt contained up to ten verifiable instructions. Through experiments with models like GPT-4o, Claude-3.5, and others, the paper identifies a phenomenon termed the "curse of instructions", where the success rate of following all instructions decreases with the number of instructions. They also proposed a method to enhance instruction-following performance through inference-time self-refinement.

The paper has received mixed feedback from the reviewers, with all reviewers agreeing on the significance of the research question but raising concerns about the contributions' originality, scope, and empirical robustness.

Strengths:
The paper addresses an important limitation in LLMs' instruction-following abilities.
ManyIFEval provides a systematic way to evaluate LLMs under increasing instruction complexity.
The proposed inference-time strategy shows improvements on multi-instruct following without retraining.

Weaknesses:
Reviewers raised concerns that ManyIFEval is a straightforward extension of existing benchmarks like IFEval or ComplexBench, limiting the paper's originality.
The dataset focuses on simple, binary-verifiable instructions which are very different from real-world LLM tasks.
The self-refinement method is not novel, as it builds on known techniques like chain-of-thought reasoning.

The recommendation is to reject the submission in its current form. While the paper identifies a valuable research problem and provides meaningful insights, the concerns raised by reviewers regarding the originality, dataset scope, and generalizability make it hard to recommend acceptance.

**Additional Comments On Reviewer Discussion:**

Novelty Concerns (aUoj, yjTk, k32n):

ManyIFEval is seen as a trivial extension of IFEval, lacking significant novelty.
The dataset construction heavily influences the results, particularly the discovered "rule" relating prompt-level accuracy to instruction-level accuracy.

Author highlighted key differences from ComplexBench, emphasizing that ManyIFEval focuses on scaling the number of instructions rather than nesting depth. Author also expanded the dataset size to improve reliability.

The novelty of ManyIFEval remains modest, as the reviewers argued that the "curse of instructions" was predictable (which I also agree).
The dataset expansion is a positive change, but it does not fully address concerns.

---

### Decision · Program_Chairs · 2025-01-22

Reject